# Evaluation of lipid biomarkers as proxies for sea ice and ocean temperatures along the Antarctic continental margin

Nele Lamping[1], Juliane Müller[1,2,3], Jens Hefter[1], Gesine Mollenhauer[1,2,3], Christian Haas[1], Xiaoxu Shi[1], Maria-Elena Vorrath[1], Gerrit Lohmann[1,3,4], Claus-Dieter Hillenbrand[5]

[1]Alfred Wegener Institute, Helmholtz Center for Polar and Marine Research, Am Alten Hafen 26, 27568 Bremerhaven, Germany

[2]Department of Geosciences, University of Bremen, Klagenfurter Straße, 28359 Bremen, Germany

[3]Marum - Center for Marine Environmental Sciences, Leobener Straße 8, 28359 Bremen, Germany

[4]Department of Environmental Physics, University of Bremen, 28359 Bremen, Germany

[5]British Antarctic Survey, High Cross, Madingley Road, Cambridge CB3 0ET, United Kingdom

*Correspondence to: Nele Lamping (nele.lamping@awi.de)*

## Abstract

The importance of Antarctic sea ice and Southern Ocean warming has come into the focus of polar research in the last couple of decades. Especially in West Antarctica, where warm water masses approach the continent and where sea ice has declined, the distribution and evolution of sea ice play a critical role for the stability of nearby ice shelves. Organic geochemical analyses of marine seafloor surface sediments from the Antarctic continental margin permit an evaluation of the applicability of biomarker-based sea ice and ocean temperature reconstructions in these vulnerable areas. We analysed highly branched isoprenoids (HBIs), such as the sea-ice proxy $IPSO_{25}$ and phytoplankton-derived HBI-trienes, but also phytosterols and isoprenoidal glycerol dialkyl glycerol tetraethers (GDGTs), which are established tools for the assessment of primary productivity and ocean temperatures, respectively. The combination of $IPSO_{25}$ with a phytoplankton marker (*i.e.* the $PIPSO_{25}$ index) permits semi-quantitative sea ice reconstructions and avoids misleading over- or underestimations of sea-ice cover. Comparisons

of the PIPSO$_{25}$-based sea-ice distribution patterns and TEX$^L_{86}$- and RI-OH'-derived ocean temperatures with (1) sea-ice concentrations obtained from satellite observations and (2) instrumental sea surface and subsurface temperatures corroborate the general capability of these proxies to properly display oceanic key variables. This is further supported by model data. We also highlight specific aspects and limitations that need to be considered when interpreting such biomarker data and discuss the potential of IPSO$_{25}$ to reflect the former occurrence of platelet ice and/or the export of ice shelf water.

# 1. Introduction

One of the key components of the global climate system, influencing major atmospheric and oceanic processes, is floating on the ocean's surface at high latitudes – sea ice (Thomas, 2017). Southern Ocean sea ice is one of the most strongly changing features of the Earth's surface as it experiences considerable seasonal variabilities with decreasing sea-ice extent from a maximum of 20 x $10^6$ km$^2$ in September to a minimum of 4 x $10^6$ km$^2$ in March (Arrigo et al., 1997; Zwally, 1983). This seasonal waxing and waning of sea ice substantially modifies deep-water formation as well as the ocean-atmosphere exchange of heat and gas, strongly affects surface albedo and radiation budgets (Abernathey et al., 2016; Nicholls et al., 2009; Turner et al., 2017), and also regulates ocean buoyancy flux, upwelling and primary production (Schofield et al., 2018).

Based on the 40-year satellite record, Southern Ocean sea-ice extent as a whole followed an increasing trend (Comiso et al., 2017; Parkinson and Cavalieri, 2012), experiencing an abrupt reversal from 2014 to 2018 (Parkinson, 2019; Turner et al., 2020; Wang et al., 2019), which has been attributed to a decades-long oceanic warming and increased advection of atmospheric heat (Eayrs et al., 2021). In particular, the sea-ice extent around major parts of West Antarctica has been decreasing (Parkinson and Cavalieri, 2012), with the Antarctic Peninsula being affected by a significant reduction in sea-ice extent and rapid atmospheric and oceanic warming (Etourneau et al., 2019; Li et al., 2014; Massom et al., 2018; Vaughan et al., 2003). The Larsen Ice Shelves A and B, located east of the Antarctic Peninsula, collapsed in 1995 and 2002, respectively, which was triggered by the loss of a sea-ice buffer, enabling an increased flexure of the ice shelf margins by ocean swells (Massom et al., 2018). The Bellingshausen and Amundsen Seas are also affected by a major sea-ice decline and regional surface ocean warming (Hobbs et al., 2016; Parkinson, 2019). Marine-terminating glaciers draining into the Amundsen Sea are thinning at an alarming rate, which has been linked to sub-ice shelf melting caused by relatively warm Circumpolar Deep Water (CDW) incursions into sub-ice shelf cavities (*e.g.*, Jacobs et al., 2011; Khazendar et al., 2016; Nakayama et al., 2018; Rignot et al., 2019; Smith et al., 2017). The disintegration of ice shelves reduces the buttressing effect that they exert on ice grounded further upstream, which may lead to a partial collapse of the catchments of the affected glaciers, eventually

raising global sea level considerably (3.4 to 4.4 m resulting from a WAIS collapse; Fretwell et al., 2013;
Jenkins et al., 2018; Pritchard et al., 2012; Vaughan, 2008).
State-of-the-art climate models are not yet fully able to depict sea-ice seasonality and sea-ice cover,
which the 5[th] Assessment Report of the Intergovernmental Panel on Climate Change (Stocker et al.,
2013) attributes to a lack of validation efforts using proxy-based sea-ice reconstructions. Knowledge
about (paleo-)sea-ice conditions and ocean temperatures in the climate sensitive areas around the West
Antarctic Ice Sheet is hence considered as crucial for understanding past and future climate evolution.
To date, the most common proxy-based sea-ice reconstructions in the Southern Ocean are conducted
by the use of sympagic diatom assemblages, which are strongly dependent on their preservation within
the sediments (Allen et al., 2011; Armand and Leventer, 2003; Crosta et al., 1998; Esper and Gersonde,
2014; Gersonde and Zielinski, 2000; Leventer, 1998). Dissolution effects within the water column or
after deposition determine the preservation state of the small, lightly silicified microfossils and may
alter the diatom record, leading to inaccurate sea-ice reconstructions (Leventer, 1998; Zielinski et al.,
1998). Recently, the molecular remains of certain diatoms, specific organic geochemical lipids, have
emerged as a potential proxy for reconstructing past Antarctic sea-ice cover (Barbara et al., 2013;
Collins et al., 2013; Crosta et al., 2021; Denis et al., 2010; Etourneau et al., 2013; Lamping et al., 2020;
Massé et al., 2011; Vorrath et al., 2019; 2020). Specifically, a di-unsaturated highly branched isoprenoid
(HBI) alkene (HBI diene, $C_{25:2}$) has been detected in both sea-ice diatoms and sediments in the Southern
Ocean (Johns et al., 1999; Massé et al., 2011; Nichols et al., 1988), and recently the sympagic (*i.e.* living
within sea ice) tube-dwelling diatom *Berkeleya adeliensis* has been identified as producer, which
preferably proliferates in platelet ice (Belt et al., 2016; Riaux-Gobin and Poulin, 2004). However, *B.*
*adeliensis* seems rather flexible concerning its habitat, since it was also recorded in the bottom ice layer
and seems to be well adapted to changes in texture during ice melt (Riaux-Gobin et al., 2013). Belt et
al. (2016) introduced the term IPSO$_{25}$ ("Ice Proxy of the Southern Ocean with 25 carbon atoms") by
analogy to the counterpart IP$_{25}$ in the Arctic. Commonly, for a more detailed assessment of sea-ice
conditions, IP$_{25}$ in the Arctic Ocean and IPSO$_{25}$ in the Southern Ocean have been measured alongside
complementary phytoplankton-derived lipids, such as sterols and/or HBI-trienes, which are indicative
of open-water conditions (Belt and Müller, 2013; Lamping et al., 2020; Etourneau et al., 2013; Vorrath
et al., 2019; 2020). The combination of the sea-ice biomarker and a phytoplankton biomarker, the so-
called PIPSO$_{25}$ index (Vorrath et al., 2019), allows for a more quantitative differentiation of contrasting
sea-ice settings and helps to avoid misinterpretations of the absence of IPSO$_{25}$ which can result from
either a lack of sea-ice cover or a permanently thick sea-ice cover, that prevents light penetration hence
limiting ice algae growth. Recently, Lamping et al. (2020) used this approach to study changes in sea-
ice conditions during the last deglaciation of the Amundsen Sea shelf, which were likely linked to
advance and retreat phases of the Getz Ice Shelf.
Multiple mechanisms exist that can cause ice shelf instability. As previously mentioned, relatively warm
CDW is considered one of the main drivers for ice shelf thinning in the Amundsen Sea Embayment
(Nakayama et al., 2018; Jenkins and Jacobs, 2008; Rignot et al., 2019). Accordingly, changing ocean
temperatures are another crucial factor for the stability of the marine-based ice streams draining most
of the West Antarctic Ice Sheet (e.g., Colleoni et al., 2018). As for sea-ice reconstructions, organic
geochemical lipids for reconstructing ocean temperatures in high latitudes have come into focus in the
past decades, since the preservation of calcareous microfossils, which are commonly used for such
reconstructions, is very poor in polar marine sediments (e.g., Zamelczyk et al., 2012). Archaeal
isoprenoidal glycerol dialkyl glycerol tetraethers (isoGDGTs), sensitive to temperature change and
relatively resistant to degradation processes, are well-preserved in marine sediments (Huguet et al.,
2008; Schouten et al., 2013). Schouten et al. (2002) found that the number of rings in sedimentary
GDGTs is correlated with surface water temperatures and developed the first archaeal lipid
paleothermometer TEX$_{86}$, a ratio of certain GDGTs, as a sea surface temperature (SST) proxy. For polar
oceans, Kim et al. (2010) developed a more specific calibration model for temperatures below 15 °C,
TEX$^{L}_{86}$, which employs a different GDGT combination. There is an emerging consensus that GDGTs
are rather reflecting subsurface ocean temperatures (SOT) along the Antarctic margin (Kim et al., 2012;
Etourneau et al., 2019; Liu et al., 2020). This is supported by observations of elevated archaeal
abundances (and GDGTs) in warmer subsurface waters (Liu et al., 2020; Spencer-Jones et al., 2021).
Archaea adapt their membrane in cold waters by adding hydroxyl groups and changing the number of
rings, OH-GDGTs (Fietz et al., 2020). The additional hydroxyl moieties lead to an increase of the
membrane fluidity that aids trans-membrane transport in cold environments, which Huguet et al. (2017)
found in molecular dynamic simulations, explaining the higher relative abundance of OH Archaea lipids
in cold environments. Taking the OH-GDGTs into account, Lü et al. (2015) proposed an SST-proxy for
the polar oceans, the RI-OH'.
Our aim with this study is to provide insight into the application of biomarkers for sea ice as well as
ocean temperature reconstructions in Southern Ocean sediments. Estimates on recent sea-ice coverage
and ocean temperatures along the eastern and western Antarctic Peninsula (EAP and WAP) as well as
in the Amundsen and Weddell Seas, are based on the analyses of $IPSO_{25}$, HBI-trienes and phytosterols
as well as GDGTs in seafloor surface sediment samples from these areas. An intercomparison of
biomarker-based sea ice as well as ocean temperature estimates with (1) sea-ice distributions obtained
from satellite observations and (2) ocean temperatures deduced from instrumental data allows for an
evaluation of the proxy approaches. We further consider AWI-ESM2 climate model data to assess the
model's performance in depicting recent oceanic key variables and to examine the potential impact of
paleoclimate conditions on the biomarker composition of the investigated surface sediments. In regard
of the various factors affecting the use of marine biomarkers as paleoenvironmental proxies, we further
comment on the limitations of GDGT temperature estimates and the novel $PIPSO_{25}$ approach, and we
discuss the potential connection between $IPSO_{25}$ and platelet ice formation under near-coastal fast ice,
which is related to the near-surface presence of sub-ice shelf melt water.

**2.   Regional setting**

The areas of investigation in this study include the southern Drake Passage, the continental shelves of
the WAP and EAP (~60° S) and the more southerly located Amundsen and Weddell Seas (~75° S; Fig.
1). The different study areas are all connected by the Antarctic Circumpolar Current (ACC), the
Antarctic Coastal Current and the Weddell Gyre (Meredith et al., 2011; Rintoul et al., 2001).

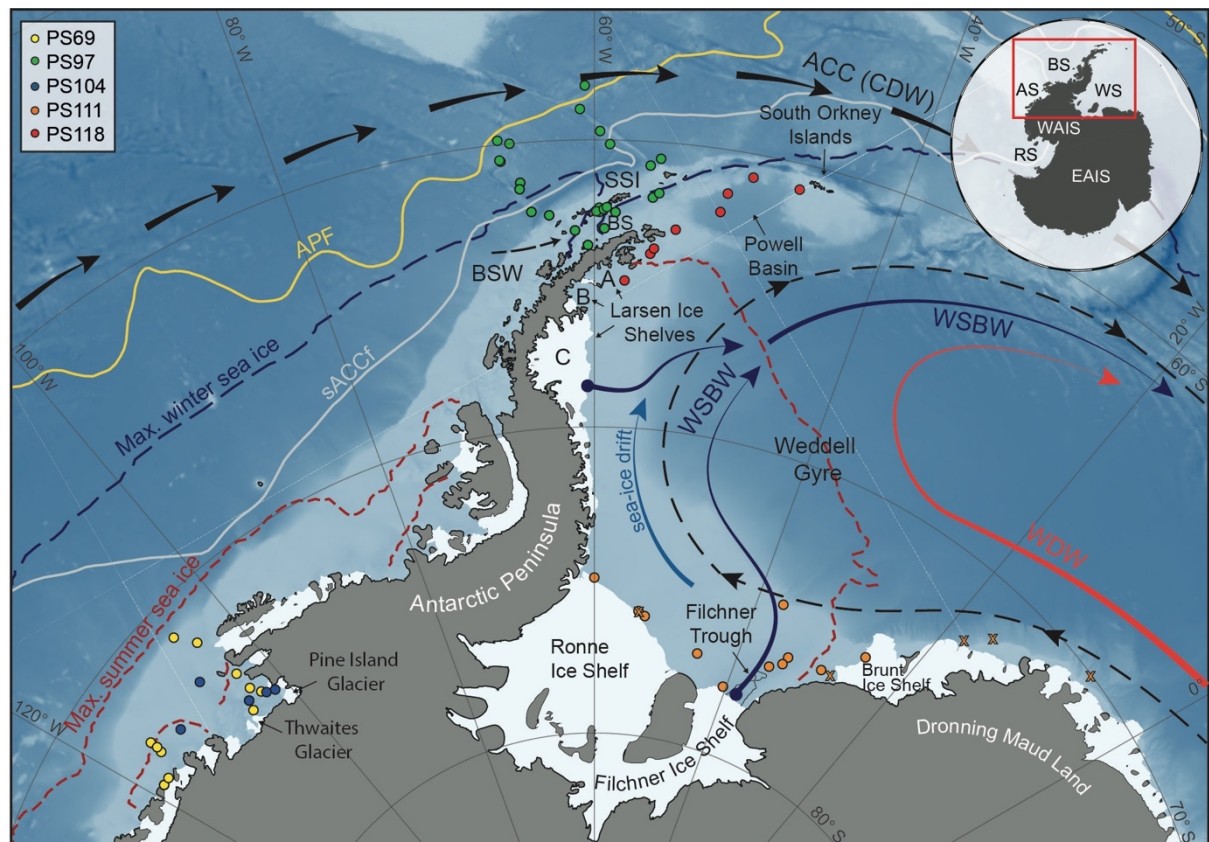

**Fig. 1:** Map of the study area (location indicated by red box in insert map) including all 41 sample locations (see different colored dots for individual *RV Polarstern* expeditions in the top left corner; for detailed sample information, see Table S1) and main oceanographic features. Maximum summer and winter sea-ice boundaries are marked by dashed red and blue line, respectively (Fetterer et al., 2016). Orange crosses indicate samples where a PIPSO$_{25}$ value of 1 has been assigned due to low biomarker concentrations, close to detection limit. ACC: Antarctic Circumpolar Current, APF: Antarctic Polar Front, sACCf: southern Antarctic Circumpolar Current Front, SSI: South Shetland Islands, BS: Bransfield Strait, BSW: Bellingshausen Sea Water, CDW: Circumpolar Deep Water; WDW: Weddell Deep Water, WSBW: Weddell Sea Bottom Water (Mathiot et al., 2011; Orsi et al., 1995). Insert map shows grounded ice only (i.e., no ice shelves), WAIS: West Antarctic Ice Sheet, EAIS: East Antarctic Ice Sheet, RS: Ross Sea, AS: Amundsen Sea, BS: Bellingshausen Sea, WS: Weddell Sea. Background bathymetry derived from IBCSO data (Arndt et al., 2013).

The ACC is mainly composed of CDW and is the largest current system in the world characterised by
a strong eastward flow, which finds its narrowest constriction in the Drake Passage. Along the
Bellingshausen Sea, the Amundsen Sea and WAP, where the ACC flows close to the continental shelf
edge, CDW is upwelling onto the shelf and flows to the coast via bathymetric troughs, contributing to
basal melt and retreat of marine-terminating glaciers and ice shelves (Cook et al., 2016; Jacobs et al.,
2011; Jenkins and Jacobs, 2008; Klinck et al., 2004). In the Weddell Sea, a subpolar cyclonic circulation
is present south of the ACC, the Weddell Gyre, which deflects part of the ACC's CDW towards the
south turning it into Warm Deep Water (WDW; Fig. 1; Hellmer et al., 2016; Vernet et al., 2019). In
close vicinity to the Filchner-Ronne and Larsen Ice Shelves, glacially derived freshwater as well as
dense brine released during sea-ice formation contribute to Weddell Sea Bottom Water (WSBW) - a
major precursor of Antarctic Bottom Water (Hellmer et al., 2016). Wind and currents force a northward
sea-ice drift in the western Weddell Sea along the eastern coast of the Antarctic Peninsula (Harms et
al., 2001) until leaving it to melt in warmer waters to the North and up to the Powell Basin (Vernet et
al., 2019). At the northern tip of the Antarctic Peninsula, colder and saltier Weddell Sea water masses
branch off westwards into the Bransfield Strait where they encounter the well-stratified, warm, and
fresh Bellingshausen Sea Water (BSW; Fig. 1), which is entering the Bransfield Strait from the West
(Sangrà et al., 2011).
Since 1978, satellite observations show strong seasonal as well as decadal changes in sea-ice cover at
the Antarctic Peninsula, which are less pronounced in the more southerly Amundsen and Weddell Seas
(Fig. 2a-c). Mean monthly sea-ice concentrations (SIC) for winter (JJA), spring (SON) and summer
(DJF) reveal a permanently ice-free Drake Passage, while the WAP and EAP shelf areas are influenced
by a changing sea-ice cover in the course of a year (Fig. 2a-c). For the Amundsen and Weddell Seas,
satellite data reveal a closed seasonal sea-ice cover with up to ~90 % concentration during winter and
spring (Fig. 2a+b), and a late break-up of sea-ice cover to a minimum concentration of ~30 % during
summer (Fig. 2c).

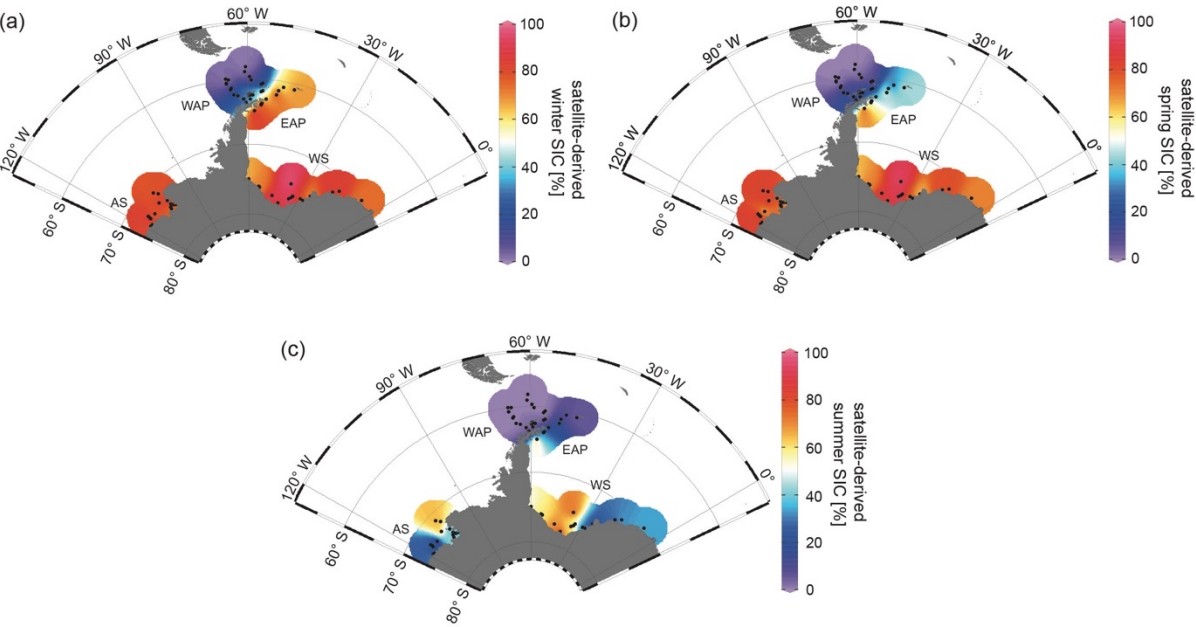

**Fig. 2:** Distribution of mean monthly satellite-derived sea-ice concentrations for (a) winter (JJA), (b) spring (SON) and (c) summer (DJF) in % (downloaded from the National Snow and Ice Data Center, NSIDC; Cavalieri et al., 1996). AS: Amundsen Sea, WAP: West Antarctic Peninsula, EAP: East Antarctic Peninsula, WS: Weddell Sea.

## 3. Material and methods

### 3.1 Sediment samples

In total, we analysed a set of 41 surface sediment samples from different areas of the Southern Ocean (Fig. 1) retrieved by multicorers and giant box corers during *RV Polarstern* expeditions over the past 15 years. Sixteen surface sediment samples from the Amundsen Sea continental shelf were collected during *RV Polarstern* expeditions PS69 in 2006 (Gohl, 2007) and PS104 in 2017 (Gohl, 2017). Twenty-five surface sediment samples from the southeastern and southwestern Weddell Sea continental shelf were collected during *RV Polarstern* expeditions PS111 in 2018 (Schröder, 2018) and PS118 in 2019 (Dorschel, 2019). This set of samples was complemented by 26 surface sediment samples from the Bransfield Strait/WAP for which the analytical results had been previously published by Vorrath et al. (2019).

### 3.2 Bulk sediment and organic geochemical analyses

The sediment material was freeze-dried and homogenized with an agate mortar and stored in glass vials at -20 °C before and after these initial preparation steps to avoid degradation of targeted molecular components. Total organic carbon (TOC) contents were measured on 0.1 g of sediment after removing inorganic carbon (total inorganic carbon, carbonates) with 500 µl 12 N hydrochloric acid. Measurements were conducted by means of a carbon-sulphur determinator (CS 2000; Eltra) with standards being measured for calibration before sample analyses and after every tenth sample to ensure accuracy (error ± 0.02 %).

Lipid biomarkers were extracted from the sediments (4 g for PS69 and PS104; 6 g for PS111 and PS118) by ultrasonication (3 x 15 min), using dichloromethane:methanol (3 x 6 ml for PS69 and PS104; 3 x 8 ml for PS111 and PS118; 2:1 v/v) as solvent. Prior to this step, the internal standards 7-hexylnonadecane (7-HND; 0.038 µg/sample for PS69 and PS104 and 0.057 µg/sample for PS111 and PS118), 5α-androstan-3-ol (1.04 µg/sample) and $C_{46}$ (0.98 µg/sample) were added to the sample for quantification of HBIs, sterols and GDGTs, respectively. Via open-column chromatography, with $SiO_2$ as stationary phase, fractionation of the extract was achieved by eluting the apolar fraction (HBIs) and the polar

fraction (sterols and GDGTs) with 5 ml n-hexane and 5 ml DCM/MeOH 1:1, respectively. The polar
fraction was subsequently split into two fractions (sterols and GDGTs) for further processing. The sterol
fraction was silylated with 300 µl bis-trimethylsilyl-trifluoroacetamide (BSTFA; 2h at 60 °C).
Compound analyses of HBIs and sterols were carried out on an Agilent Technologies 7890B gas
chromatograph (GC; fitted with a 30 m DB 1MS column; 0.25 mm diameter and 0.25 µm film thickness)
coupled to an Agilent Technologies 5977B mass selective detector (MSD; with 70 eV constant
ionization potential, ion source temperature of 230 °C). The GC oven was set to: 60 °C (3 min), 150 °C
(rate: 15 °C/min), 320 °C (rate: 10 °C/min), 320 °C (15 min isothermal) for the analysis of hydrocarbons
and to: 60 °C (2 min), 150 °C (rate: 15 °C/min), 320 °C (rate: 3 °C/min), 320 °C (20 min isothermal)
for the analysis of sterols. Helium was used as carrier gas. The identification of HBI and sterol
compounds is based upon their GC retention times and mass spectra (Belt, 2018; Belt et al., 2000; Boon
et al., 1979). Lipid quantification was obtained by setting the individual, manually integrated, GC-MS
peak area in relation to the peak area of the respective internal standard and normalization to the amount
of extracted sediment. Quantification of $IPSO_{25}$ and HBI trienes was achieved using their molecular
ions ($IPSO_{25}$: m/z 348 and HBI trienes: m/z 346) in relation to the fragment ion m/z 266 of the internal
standard 7-HND (Belt, 2018). Quantification of sterols was achieved by comparison of the molecular
ion of the individual sterol with the molecular ion m/z 348 of the internal standard 5α-androstan-3-ol.
Instrumental response factors for the target lipids were considered as recommended by Belt et al. (2014)
and Fahl and Stein (2012). All biomarker concentrations were subsequently normalized to the TOC
content of each sample to account for different depositional settings within the different study areas.
For calculating the phytoplankton-$IPSO_{25}$ ($PIPSO_{25}$) index, we used the equation introduced by Vorrath
et al. (2019):
$PIPSO_{25} = IPSO_{25} / (IPSO_{25} + (\text{phytoplankton marker x c}))$ (1)
where c (c = mean $IPSO_{25}$/mean phytoplankton marker) is applied as a concentration balance factor to
account for high concentration offsets between $IPSO_{25}$ and the phytoplankton biomarker (see Table S1
for c-factors of individual $PIPSO_{25}$ calculations).
Following the approach by Müller and Stein (2014) and Lamping et al. (2020), samples with
exceptionally low (at detection limit) concentrations of both biomarkers have been assigned a $PIPSO_{25}$
value of 1 (see chapter 4.1.2). This comprises the five Weddell Seam samples PS111/13-2, /15-1, /16-
3, /29-3 and /40-2 (marked as orange x in Fig. 1).
The GDGT fraction was dried under $N_2$, redissolved with 120 µl hexane:isopropanol (v/v 99:1) and
then filtered using a polytetrafluoroethylene (PTFE) filter with a 0.45 µm pore sized membrane. GDGT
measurements were carried out using high performance liquid chromatography (HPLC; Agilent 1200
series HPLC system) coupled to an Agilent 6120 mass spectrometer (MS), operating with atmospheric
pressure chemical ionization (APCI). The injection volume was 20 µl. For separating the GDGTs, a
Prevail Cyano 3 µm column (Grace, 150 mm * 2.1 mm) was kept at 30 °C. Each sample was eluted
isocratically for 5 min with solvent A = hexane/2-propanol/chloroform; 98:1:1 at a flow rate of 0.2
ml/min, then the volume of solvent B = hexane/2-propanol/chloroform; 89:10:1 was increased linearly
to 10 % within 20 min and then to 100 % within 10 min. The column was back-flushed (5 min, flow
0.6 ml/min) after 7 min after each sample and re-equilibrated with solvent A (10 min, flow 0.2 ml/min).
The APCI was set to the following: $N_2$ drying gas flow at 5 l/min and temperature to 350 °C, nebulizer
pressure to 50 psi, vaporizer gas temperature to 350 °C, capillary voltage to 4 kV and corona current to
+5 µA. Detection of GDGTs was achieved by means of selective ion monitoring (SIM) of $[M+H]^+$ ions
(dwell time 76 ms). Determination and quantification of the molecular ions of GDGT-0 (*m/z* 1302),
GDGT-1 (*m/z* 1300), GDGT-2 (*m/z* 1298), GDGT-3 (*m/z* 1296) and crenarchaeol (*m/z* 1292) as well as
of brGDGT-III (*m/z* 1050), brGDGT-II (*m/z* 1036) and brGDGT-I (*m/z* 1022) was done in relation to
the molecular ion *m/z* 744 of the internal standard $C_{46}$-GDGT. The late eluting hydroxylated GDGTs
(OH-GDGT-0, OH-GDGT-1 and OH-GDGT-2 with *m/z* 1318, 1316 and 1314, respectively) were
quantified in the scans (*m/z* 1300, 1298, 1296) of their related GDGTs, as described by Fietz et al.

(2013).

$TEX^L_{86}$ values and their conversion into SOTs were determined following Kim et al. (2012):
$$TEX^L_{86} = LOG \frac{[GDGT-2]}{[GDGT-1]+[GDGT-2]+[GDGT-2]} , \tag{2}$$
$$SOT^{TEX} [°C] = 50.8 \times TEX^L_{86} + 36.1. \tag{3}$$
Temperature calculations based on OH-GDGTs were carried out according to Lü et al. (2015):
$$RI-OH' = \frac{[OH-GDGT-1]+2 \times [OH-GDGT-2]}{[OH-GDGT-0]+[OH-GDGT-1]+[OH-GDGT-2]} , \tag{4}$$
$SST^{OH} \ [ \ ^{\circ}C] = RI - OH' - 0.1/0.0382$ .                                (5)
To determine the relative influence of terrestrial organic matter input, the Branched Isoprenoid
Tetraether (BIT)-index was calculated following Hopmans et al. (2004):
$BIT = \dfrac{[brGDGT-I]+[brGDGT-II]+[brGDGT-III]}{[Chrenarchaeol]+[brGDGT-I]+[brGDGT-II]+[brGDGT-III]}$ .             (6)

3.3     Numerical model

3.3.1   Model description

AWI-ESM2 is a state-of-the-art coupled climate model developed by Sidorenko et al. (2019) which
comprises an atmospheric component ECHAM6 (Stevens et al., 2013) as well as an ocean-sea ice
component FESOM2 (Danilov et al., 2017). The atmospheric module ECHAM6 is the most recent
version of the ECHAM model developed at the Max Planck Institute for Meteorology (MPI) in
Hamburg. The model is branched from an early release of the European Center (EC) for Medium Range
Weather Forecasts (ECMWF) model (Roeckner et al., 1989). ECHAM6 dynamics is based on
hydrostatic primitive equations with traditional approximation. We used a T63 Gaussian grid which has
a spatial resolution of about 1.9 x 1.9 degree (1.9 ° or 210 km). There are 47 vertical layers in the
atmosphere.
Momentum transport arising from boundary effects is configured using the subgrid orography scheme
as described by Lott (1999). Radiative transfer in ECHAM6 is represented by the method described in
Iacono et al. (2008). ECHAM6 also contains a Land-Surface Model (JSBACH) which includes 12
functional plant types of dynamic vegetation and 2 bare-surface types (Loveland et al., 2000; Raddatz
et al., 2007). The ice-ocean module in AWI-ESM2 is based on the finite volume discretization
formulated on unstructured meshes. The multi-resolution for the ocean is up to 15 km over polar and
coastal regions, and 135 km for far-field oceans, with 46 uneven vertical depths. The impact of local
dynamics on the global ocean is related to a number of FESOM-based studies (Danilov et al., 2017).
The multi-resolution approach advocated by FESOM allows one to explore the impact of local
processes on the global ocean with moderate computational effort (Danilov et al., 2017). AWI-ESM2
employs the OASIS3-MCT coupler (Valcke, 2013) with an intermediate regular exchange grid.

Mapping between the intermediate grid and the atmospheric/oceanic grid is handled with bilinear interpolation. The atmosphere component computes 12 air–sea fluxes based on four surface fields provided by the ocean module FESOM2. AWI-ESM2 has been validated under modern climate conditions (Sidorenko et al., 2019) and has been applied for marine radiocarbon concentrations (Lohmann et al., 2020), the latest Holocene (Vorrath et al., 2020), and the Last Interglacial (Otto-Bliesner et al., 2021).

### 3.3.2  Experimental design

One transient experiment was conducted using AWI-ESM2, which applied the boundary conditions, including orbital parameters and greenhouse gases. Orbital parameters are calculated according to Berger (1978), and the concentrations of greenhouse gases are taken from ice-core records as well as from recent measurements of firn air and atmospheric samples (Köhler et al., 2017). The model was initialized from a 1,000-year spin-up run under mid-Holocene (6,000 before present, BP) boundary conditions as described by Otto-Bliesner et al. (2017). In our modeling strategy, we follow Lorenz and Lohmann (2004) and use the climate condition from the mid-Holocene spin-up run as the initial state for the subsequent transient simulation covering the period from 6,000 BP to 2014 CE. In the present study we derived seasonal SIC, SSTs and SOTs in the study area from a segment of the transient experiment (1950-2014 CE). Topography including prescribed ice sheet configuration was kept constant in our transient simulation. All model data are provided in Table S2.

### 3.4.  Satellite SIC and SSTs

Satellite sea-ice data are derived from Nimbus-7 SMMR and DMSP SSM/I-SSMIS passive microwave data and downloaded from the National Snow and Ice Data Center (NSIDC; Cavalieri et al., 1996). The sea-ice data represent mean monthly SIC, which are expressed to range from 0 % to 100 % and are averaged over a period of the beginning of satellite observations in 1978 CE to the individual year of sample retrieval. The monthly mean SIC were then split into different seasons: winter (JJF), spring (SON) and summer (DJF) (Fig. 2a-c) and the data are considered to represent the recent mean state of sea-ice coverage. All satellite data are provided in Table S3.

Modern annual mean SSTs and SOTs are derived from the World Ocean Atlas 13 representing averaged
values for the years 1955-2012 CE (WOA13; Locarnini et al., 2013).

**4. Results and discussion**
In the following, we first present and discuss the biomarker data assembled during this study from North
(Antarctic Peninsula) to South (Amundsen and Weddell Seas) and draw conclusions about the
environmental settings deduced from the data set. As phytoplankton-derived biomarkers, we here focus
on the significance of HBI Z-triene and brassicasterol, while HBI E-triene and dinosterol - showing
very similar patterns - are moved to the supplement (Fig. S1) to avoid repetition. All biomarker data
collected during this study are provided in Table S1 and are available via the PANGAEA data repository
(in prep.). For the discussion of the target environmental variables, *i.e.* PIPSO$_{25}$-based sea ice and
GDGT-derived ocean temperature estimates, satellite and instrumental as well as modelled data are
considered. In Sect. 5, we further address potential caveats in biomarker-based environmental
reconstructions that need to be considered when applying these proxies.

4.1 TOC content, HBIs and sterols in Antarctic surface sediments
TOC contents in marine sediments in a first approximation are often viewed as an indicator for primary
productivity in surface waters (Meyers, 1997). However, we are aware that additional factors, such as
different water depths or depositional regimes, may exert control on sedimentary TOC as well. The
TOC contents of the herein investigated surface samples are lowest in Drake Passage with values around
0.12-0.54 %, increasing in a northwest-southeast gradient into Bransfield Strait, ranging between 0.59
and 1.06 % (Fig. 3a; WAP). At the EAP, higher TOC contents (0.57-0.86 %) prevail around the Larsen
Ice Shelf with a decreasing trend towards the Powell Basin (0.22-0.37 %) and an increase to 0.50 %
around the area of the South Orkney Islands, which may point to elevated productivity or enhanced
supply of reworked terrigenous organic matter in these areas (Fig. 3a; EAP).

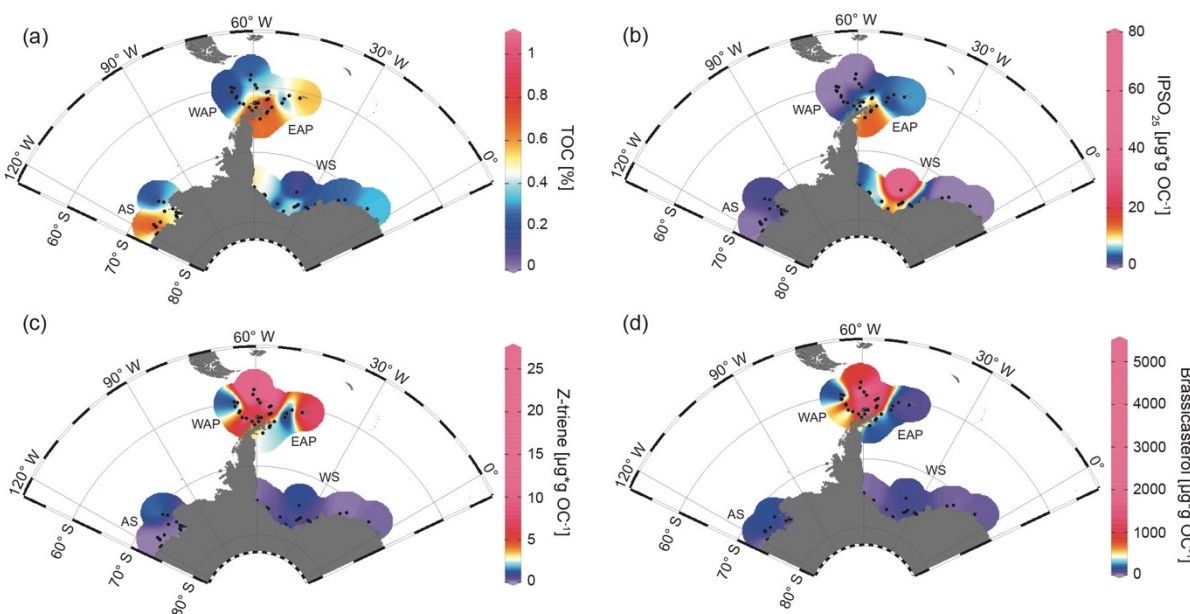

**Fig. 3:** Distribution of (a) TOC [%], (b) IPSO$_{25}$, (c) Z-triene and (d) brassicasterol in surface sediment samples. Sample locations are marked as black dots. Concentrations of biomarkers [µg*g OC$^{-1}$] were normalized to the TOC content of each sample. AS: Amundsen Sea, WAP: West Antarctic Peninsula, EAP: East Antarctic Peninsula, WS: Weddell Sea.

At the WAP, concentrations of the sea-ice biomarker IPSO$_{25}$ show a northwest-southeast gradient with
IPSO$_{25}$ being absent in samples from the permanently ice-free Drake Passage and increasing
concentrations towards the continental slope and the seasonally ice-covered continental shelf (0.37-
17.81 µg*g OC$^{-1}$; Fig. 3b; Vorrath et al., 2019). Highest IPSO$_{25}$ concentrations are observed in samples
of the northern Bransfield Strait, which is affected by inflow of water masses from the Weddell Sea
through the Antarctic Sound and along the Antarctic Peninsula and frequent transport of sea ice into the
Bransfield Strait (Vorrath et al., 2019). Elevated IPSO$_{25}$ concentrations are also observed at the EAP,
influenced by a seasonal sea-ice cover, where relatively higher concentrations of the sea-ice biomarker
prevail in those samples located in front of the Larsen Ice Shelf (12.59-17.74 µg*g OC$^{-1}$; Fig. 3b). As
these locations are also influenced by the northward drift of sea ice within the Weddell Gyre (Fig. 1),
the elevated IPSO$_{25}$ concentrations could also result from sea ice advected from the southern Weddell
Sea. We suggest that the decreasing IPSO$_{25}$ concentrations towards the Powell Basin and the South
Orkney Islands (0.59-5.36 µg*g OC$^{-1}$; Fig. 3b) can be connected to warmer ocean temperatures towards
the North and less sea-ice coverage during spring.
Concentrations of the phytoplankton biomarker HBI Z-triene around the Antarctic Peninsula are highest
in the eastern Drake Passage and along the continental slope (where $IPSO_{25}$ is absent) and decrease in
the Bransfield Strait (0.33-26.86 µg*g $OC^{-1}$; Fig. 3c; Vorrath et al., 2019). Elevated HBI Z-triene
concentrations have thus far been detected in surface waters along an ice edge (Smik et al., 2016) and
hence suggested to be a proxy for marginal ice zone conditions (Belt et al., 2015; Collins et al., 2013;
Schmidt et al., 2018). Vorrath et al. (2019), however, relate the high concentrations of HBI Z-triene at
the northernmost stations in the permanently ice-free eastern Drake Passage to their proximity to the
Antarctic Polar Front. Here, productivity of the source diatoms of HBI-trienes (*e.g.*, *Rhizosolenia* spp.;
Belt et al., 2017) may be enhanced by meander-induced upwelling leading to increased nutrient flux to
surface waters (Moore and Abbott, 2002). Since Cardenas et al. (2019) document only minor
abundances of *Rhizosolenia* spp. in surface sediments from this area, we assume that HBI-trienes might
also be biosynthesized by other diatoms. Moderate concentrations along the continental slope of the
WAP and in the Bransfield Strait have been associated with elevated inflow of warm BSW which leads
to a retreating sea-ice margin during spring and summer (for more details, see Vorrath et al., 2019;
2020). Samples from the EAP continental shelf and the Powell Basin are characterised by relatively
low concentrations of HBI Z-triene (Fig. 3c; 0.1-2.37 µg*g $OC^{-1}$), showing a southwest-northeast
gradient, while the northernmost sample closest to the South Orkney Islands is characterized by an
elevated HBI Z-triene concentration of ~8.49 µg*g $OC^{-1}$ (Fig. 3c; EAP). This relatively high
concentration may be related to an "Island Mass Effect", coined by Doty and Oguri (1956), which refers
to an increased primary production around oceanic islands in comparison to surrounding waters. Nolting
et al. (1991) found extraordinarily high dissolved iron levels (as high as 50-60 nM) on the shelf of the
South Orkney Islands, and Nielsdóttir et al. (2012) observed enhanced iron and Chl *a* concentrations in
the vicinity of the South Orkney Islands. These authors explain the increased dissolved iron levels with
input from seasonally retreating sea ice, which is recorded by satellites (Fig. 2a-c) and probably leads
to substantial annual phytoplankton blooms, which may also cause the elevated TOC content in the
corresponding seafloor sediment sample (Fig. 3a). Alternatively, remobilization of shelf sediments or
vertical mixing of iron-rich deep waters leading to high iron contents in surface waters may stimulate
primary productivity (Blain et al., 2007; de Jong et al., 2012).  However, it remains unclear why the
brassicasterol concentration is distinctly low in this sample, and we assume that different environmental
preferences of the source organisms may account for this. In Drake Passage and the EAP, brassicasterol
displays a similar pattern as the HBI Z-triene, with relatively high concentrations (more than 2 orders
of magnitudes), ranging between 1.86 and 5017.44 $\mu g \cdot g$ $OC^{-1}$ (Fig. 3d).
In the Weddell Sea, TOC contents are generally lower (< 0.4 %), with slightly elevated values in the
West (up to 0.50 %) and right in front of the Filchner Ice Shelf (up to 0.52 %; Fig. 3a). The Amundsen
Sea is characterized by slightly higher TOC contents, with concentrations of up to 0.91 % in the West
and lower values in the East (0.33 %; Fig. 3a; AS).
In the samples from the Amundsen and Weddell Seas, dominated by a strong winter sea-ice cover
lasting until spring (Fig. 2a-c), all three biomarkers are present in low concentrations only. An exception
are the samples located in front of the Filchner Ice Shelf with significantly higher concentrations of
$IPSO_{25}$ (7.09-73.87 $\mu g \cdot g$ $OC^{-1}$; Fig. 3b; WS). Concentrations of $IPSO_{25}$ on the Amundsen Sea shelf are
relatively low (0.04-3.3 $\mu g \cdot g$ $OC^{-1}$), with slightly higher values observed towards the north-east (Fig.
3b; AS). HBI Z-triene is also very low concentrated, showing slightly higher concentrations in Filchner
Trough (0.04-1 $\mu g \cdot g$ $OC^{-1}$) and towards the more distal locations in the northeastern Amundsen Sea
(0.01-1.88 $\mu g \cdot g$ $OC^{-1}$; Fig. 3c). Brassicasterol generally shows similar patterns as the HBI Z-triene,
with concentrations ranging between 1.86 and 220.54 $\mu g \cdot g$ $OC^{-1}$ (Fig. 3d; for HBI E-triene and
dinosterol distribution, see Fig. S1).

4.2    Combining individual biomarker records: the $PIPSO_{25}$ index

The $PIPSO_{25}$ index combines the relative concentrations of $IPSO_{25}$ and a selected phytoplankton
biomarker, such as HBI-trienes and sterols, as indicator for an open-ocean environment (Vorrath et al.,
2019). The combination of both end members (sea ice vs. open-ocean) prevents misleading
interpretations regarding the absence of $IPSO_{25}$ in the sediments, which can be the result of two entirely
different scenarios. At heavy/perennial sea-ice conditions, the thickness of sea ice hinders light
penetration, thereby limiting the productivity of algae living in basal sea ice (Hancke et al., 2018). This
scenario may cause the absence of both phytoplankton and sea-ice biomarkers in the sediment. The
other scenario depicts a permanently open ocean, where the sea-ice biomarker is absent as well, but

here, the phytoplankton biomarkers are present in variable concentrations (Müller et al., 2011). The

presence of both biomarkers in the sediment is indicative of seasonal sea-ice coverage and/or the

occurrence of stable sea-ice margin conditions, promoting biosynthesis of both biomarkers (Müller et

al., 2011). We here distinguish between $P_Z IPSO_{25}$ and $P_B IPSO_{25}$ using HBI Z-triene and brassicasterol

as phytoplankton biomarker, respectively (Fig. 4a+b; for $PIPSO_{25}$ values based on HBI E-triene and

dinosterol see Table S1 and Fig. S2).

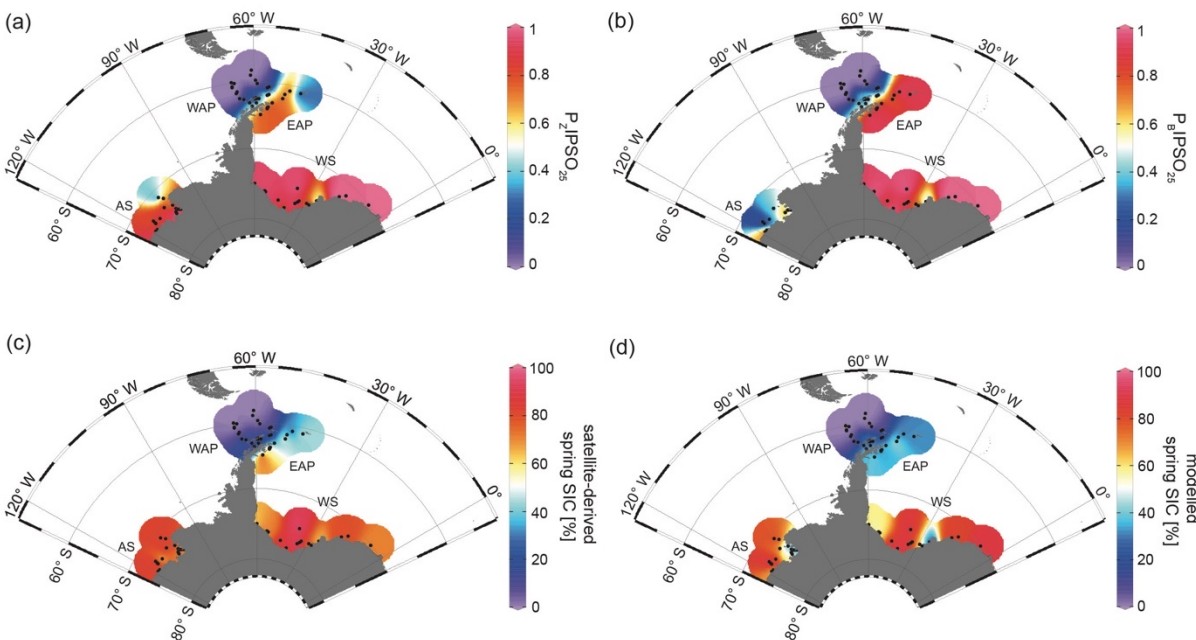

**Fig. 4:** Distribution of the sea-ice index $PIPSO_{25}$ in surface sediment samples, with (a) $P_Z IPSO_{25}$ based on Z-triene and (b) $P_B IPSO_{25}$ based on brassicasterol, (c) satellite-derived spring SIC [%] and (d) modelled spring SIC [%]. AS: Amundsen Sea, WAP: West Antarctic Peninsula, EAP: East Antarctic Peninsula, WS: Weddell Sea.

Both $PIPSO_{25}$ indices are 0 in the predominantly ice-free Drake Passage and display a northwest-

southeast gradient to intermediate values towards the continental slope and the South Shetland Islands,

reflecting increased influence of marginal sea-ice cover towards the coast (0.02-0.70; Vorrath et al.,

2019). At the seasonally sea-ice covered EAP, $P_Z IPSO_{25}$ values reach 0.84, while lower values of around

0.25 are observed close to the South Orkney Islands, which relates to the elevated HBI Z-triene

concentrations at the stations there (Fig. 3c; EAP). The $P_B IPSO_{25}$ index exhibits even higher values of

up to 0.98 at the EAP/northwestern Weddell Sea. These elevated $PIPSO_{25}$ indices align well with the

significant northward ice-drift within the Weddell Gyre in that region, which leads to prolonged sea-ice

cover along the EAP.

In samples from the southern Weddell Sea, both PIPSO$_{25}$ indices show a similar pattern with high values
up to 0.9, and slightly lower values in front of the Brunt Ice Shelf (0.6; Fig. 4a+b). Very low
concentrations (close to detection limit) of both biomarkers in samples located on the continental shelf
off Dronning Maud Land (Fig. 1) result in low PIPSO$_{25}$ values, strongly underestimating the sea-ice
cover in that area. Regarding the satellite-derived sea-ice data, this area of the continental shelf is
influenced by a severe seasonal sea-ice cover (Fig. 2). As previously mentioned, we followed the
approach by Müller and Stein (2014) and Lamping et al. (2020) and assigned a maximum PIPSO$_{25}$ value
of 1 to these samples to circumvent misleading interpretations and aid visualisation.
The intermediate PIPSO$_{25}$ value (~0.51) derived for one sample collected in front of the Brunt Ice Shelf
points to a less severe sea-ice cover in that area. A possible explanation for the relatively lower PIPSO$_{25}$
value may be the presence of a coastal polynya that has been reported by Anderson (1993) and which
is further supported by Paul et al. (2015), who note that the sea-ice area around the Brunt Ice Shelf is
the most active in the southern Weddell Sea, with an annual average polynya area of $3516 \pm 1420$ km².
Interestingly, the reduced SIC here is also captured by our model (see Sect. 4.3).
PIPSO$_{25}$ values in the Amundsen Sea point to different scenarios. The P$_Z$IPSO$_{25}$ index ranges around
0.9 with only the easterly, more distal locations showing lower values between 0.3 and 0.6 (Fig. 4a).
The P$_B$IPSO$_{25}$ index generally presents lower values ranging from 0.6 in the coastal area to 0.2 in the
more distal samples (Fig. 4b). This difference between P$_Z$IPSO$_{25}$ and P$_B$IPSO$_{25}$ may be explained by
the different source organisms biosynthesizing the individual phytoplankton biomarkers. While the
main origin of HBI-trienes seems to be restricted to diatoms (Belt et al., 2017), brassicasterol is known
to be produced by several algal groups adapted to a wider range of sea surface conditions (Volkman,
2006; see Sect. 5.2).

4.3    Biomarker-based sea ice estimates vs. satellite and model data
The main ice algae bloom in the Southern Ocean occurs during spring, when solar insolation and air
temperatures/SSTs increase and sea ice starts melting, which results in the release of nutrients and
stratification of the water column stimulating the productivity of photosynthesizing organisms (Arrigo,
2017; Belt, 2018). The sea-ice biomarker IPSO$_{25}$ is hence commonly interpreted as a spring sea-ice

indicator, which is why, in the following, we compare the biomarker-based sea-ice reconstructions to satellite-derived and modelled spring SIC. $IPSO_{25}$ concentrations in the surface sediments around the Antarctic Peninsula exhibit similar trends as the satellite-derived and modelled SIC (Figs. 3+4), while they differ significantly in the Amundsen and Weddell Seas, where high SIC are depicted by satellites and the model but $IPSO_{25}$ is very low concentrated. The low $IPSO_{25}$ concentrations in these areas highlight the uncertainty when considering $IPSO_{25}$ as a sea-ice proxy alone, since such low concentrations are not only observed under open water conditions, but also under a severe sea-ice cover. In this case, the low concentrations of $IPSO_{25}$ are the result of the latter, where limited light availability hinders ice algae growth, leading to an underestimation of sea-ice cover. Accordingly, we note a weak correlation between $IPSO_{25}$ data and satellite SIC ($R^2 = 0.19$; Fig. 5a). As stated above, the combination of $IPSO_{25}$ and a phytoplankton marker may prevent this ambiguity. The higher sea-ice concentrations in the Amundsen and Weddell Seas are better reflected by maximum $P_ZIPSO_{25}$ values than by $IPSO_{25}$ alone. However, we note that the $P_ZIPSO_{25}$ index seems to not further resolve SICs higher than 50 % (see Fig. S3), which may indicate a threshold (here ~50 % SIC) where the growth of the HBI triene and $IPSO_{25}$ producing algae is limited.

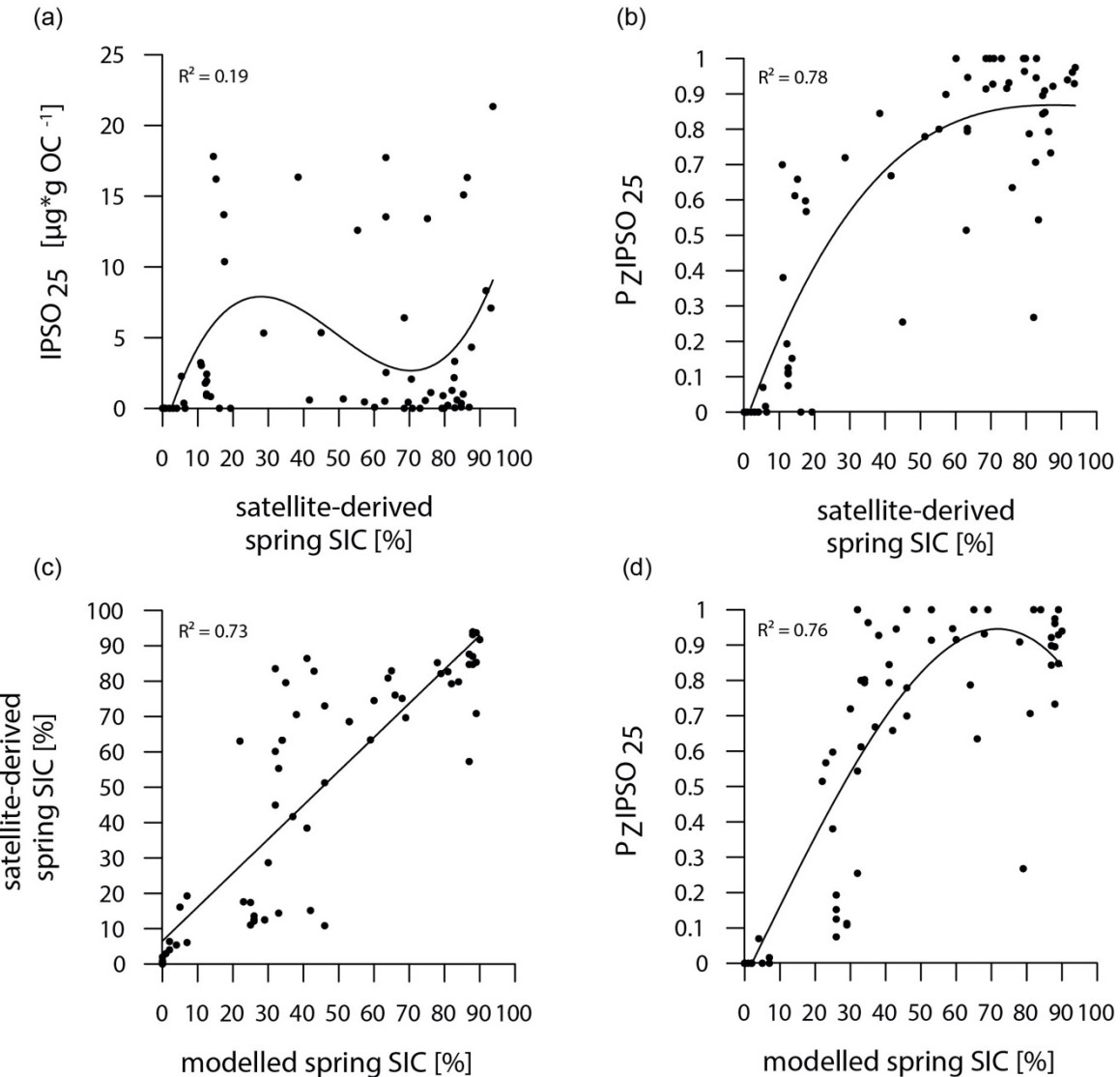

**Fig. 5:** Correlations of (a) IPSO$_{25}$ concentrations vs. satellite-derived spring SIC, (b) P$_Z$IPSO$_{25}$ values vs. satellite-derived spring SIC, (c) satellite-derived spring SIC vs. modelled spring SIC and (d) P$_Z$IPSO$_{25}$ values vs. modelled spring SIC. Coefficients of determination ($R^2$) are given for the respective regression lines.

In general, however, the P$_Z$IPSO$_{25}$ values correlate much better with satellite and modelled SIC ($R^2$ =
0.78 and $R^2$ = 0.76, respectively; Fig. 5b+d) than IPSO$_{25}$ concentrations. Correlations of satellite and
model data with PIPSO$_{25}$ calculated using the HBI E-triene, brassicasterol and dinosterol, respectively,
are also positive but less significant (Fig. S4) and we hence focus the discussion on P$_Z$IPSO$_{25}$. The AWI-
ESM2-derived spring SICs correctly display the permanently ice-free Drake Passage and the northwest-
southeast gradient in sea-ice cover from the WAP continental slope towards the Bransfield Strait (Fig.
4d). The model, however, significantly underestimates the elevated sea-ice concentrations (up to 70 %)
in front of the former Larsen Ice Shelf A and east of James Ross Island at the EAP depicted by satellite
data. In the Amundsen and Weddell Seas, the model shows a heavy sea-ice cover (~90 %), only slightly
underestimating the sea-ice cover at the near-coastal sites in front of Pine Island Glacier and the Ronne
Ice Shelf. Interestingly, modelled SIC in the area in front of the Brunt Ice Shelf is as low as ~45 % (Fig.
4d+e), corresponding well with the reduced $P_ZIPSO_{25}$ value of ~0.51, and may reflect the polynya
conditions in that region documented by Anderson (1993) and Paul et al. (2015). Overall, we note that
modelled modern SICs correlate well with satellite data ($R^2 = 0.73$; Fig. 5c) and $P_ZIPSO_{25}$ values ($R^2 =$
0.76; Fig. 5d), while we observe weaker correlations between modelled paleo-SICs and $P_ZIPSO_{25}$ values
(Fig. S5; see Sect. 5.1).

4.4   $TEX^L_{86}$- and RI-OH'- derived ocean temperatures

For a critical appraisal of the applicability and reliability of GDGT indices as temperature proxies in
polar latitudes, we here focus on the $TEX^L_{86}$ proxy by Kim et al. (2012), potentially reflecting SOTs,
and the RI-OH' proxy, assumed to reflect SSTs, by Lü et al. (2015).  The reconstructions are considered
to represent annual mean ocean temperatures (for correlations of $TEX^L_{86}$-derived SOTs with WOA
spring and winter SOTs, see Fig. S6). In all samples, the BIT-index (Eq. 6) is <0.3, indicating no
significant contribution of terrestrial input influencing the distribution and hence applicability of
GDGTs to estimate ocean temperatures. RI-OH'-derived temperatures and $TEX^L_{86}$-derived SOTs both
show a similar pattern, but different temperatures ranging between -2.62 to +4.67 °C and -2.38 to
+8.75 °C, respectively (Fig. 6a+b). At the WAP, RI-OH'- as well as $TEX^L_{86}$-derived temperatures
follow a northwest-southeast gradient with higher temperatures in the permanently ice-free Drake
Passage and on the Antarctic continental slope, influenced by the ACC and relatively warm CDW (Orsi
et al., 1995; Rintoul et al., 2001). Temperatures decrease towards the Bransfield Strait and the EAP,
which are influenced by a seasonal sea-ice cover and relatively colder and highly saline water from the
Weddell Sea, branching off the Weddell Gyre (Collares et al., 2018; Thompson et al., 2009). At the
EAP, a southwest-northeast gradient can be observed with relatively lower temperatures along the
Larsen Ice Shelf and higher temperatures towards the Powell Basin and the South Orkney Islands (Fig.
6a+b).

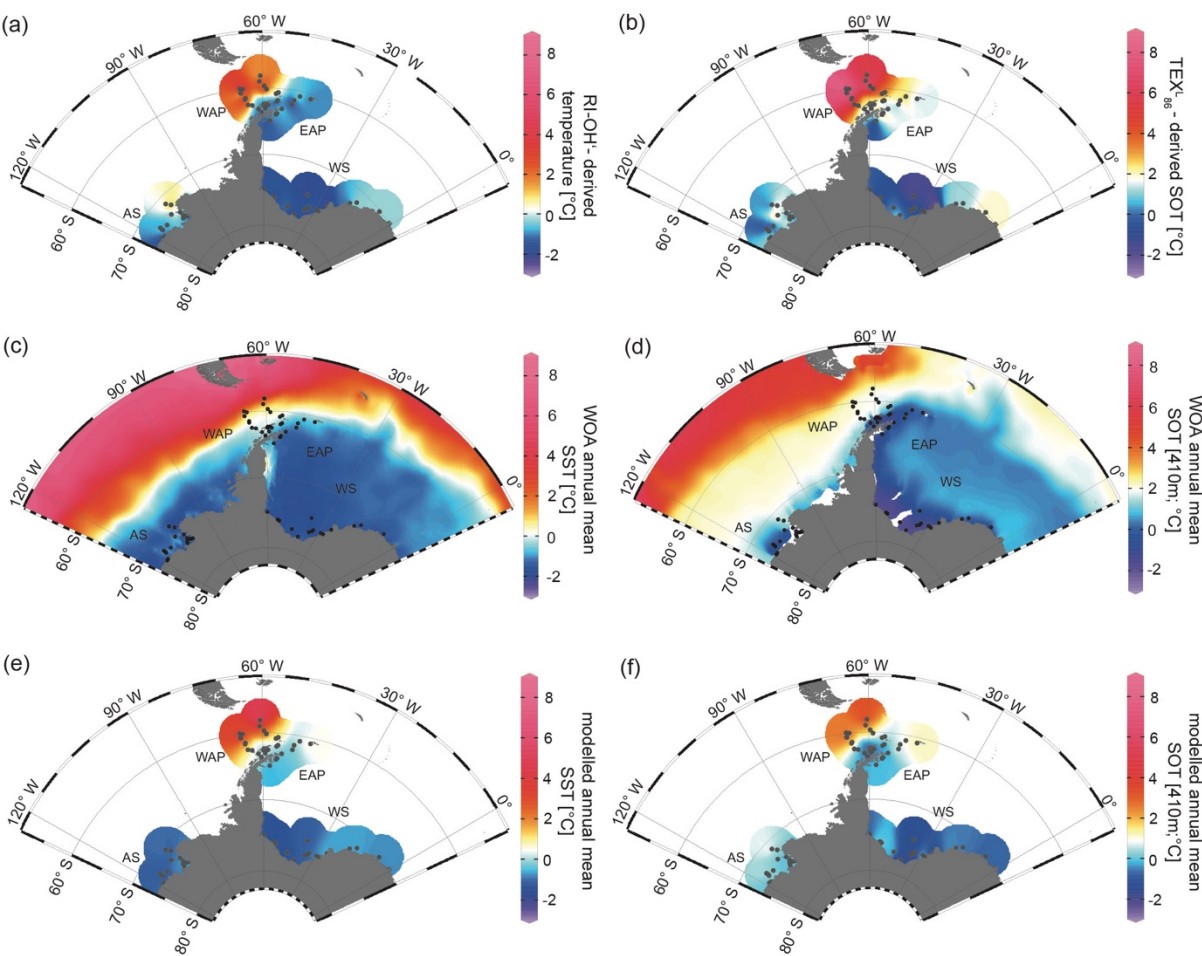

**Fig. 6:** Annual mean temperature distributions with (a) RI-OH´-derived temperature, (b) TEX$^L_{86}$-derived SOT, (c) WOA13 SST (Locarnini et al., 2013), (d) WOA13 SOT (410 m; Locarnini et al., 2013), (e) modelled SST and (f) modelled SOT (410 m) in °C. AS: Amundsen Sea, WAP: West Antarctic Peninsula, EAP: East Antarctic Peninsula, WS: Weddell Sea.

Further to the South, in the Amundsen and Weddell Seas, temperatures are generally lower than at the
Antarctic Peninsula. Samples from the Weddell Sea display a temperature decrease from east to west,
which may reflect the eddy-driven route in the north-eastern corner of the Weddell Gyre carrying
relatively warm, salty CDW, which then advects westward along the southern edge of the Weddell Gyre
as WDW (Vernet et al., 2019). Coldest TEX$^L_{86}$ as well as RI-OH' temperatures (<0 °C) at sites along
the Filchner-Ronne Ice Shelf front may be further linked to the presence of cold precursor water masses
for WSBW.
With regard to ongoing discussions, whether GDGT-based temperature reconstructions represent SSTs
or SOTs (Kalanetra et al., 2009; Kim et al., 2012; Park et al., 2019), we here compare our RI-OH' and
TEX$^L_{86}$-derived temperatures with instrumental and modelled surface as well as subsurface temperature
data (Fig. 6c-f). Based on correlations of GDGT-derived temperatures with WOA13 temperatures
reflecting different water depths, we observe the highest significance at a water depth of 410 m (for
respective correlations, see Fig. S7). When discussing instrumental and modelled SOTs, we hence refer
to 410 m water depth.

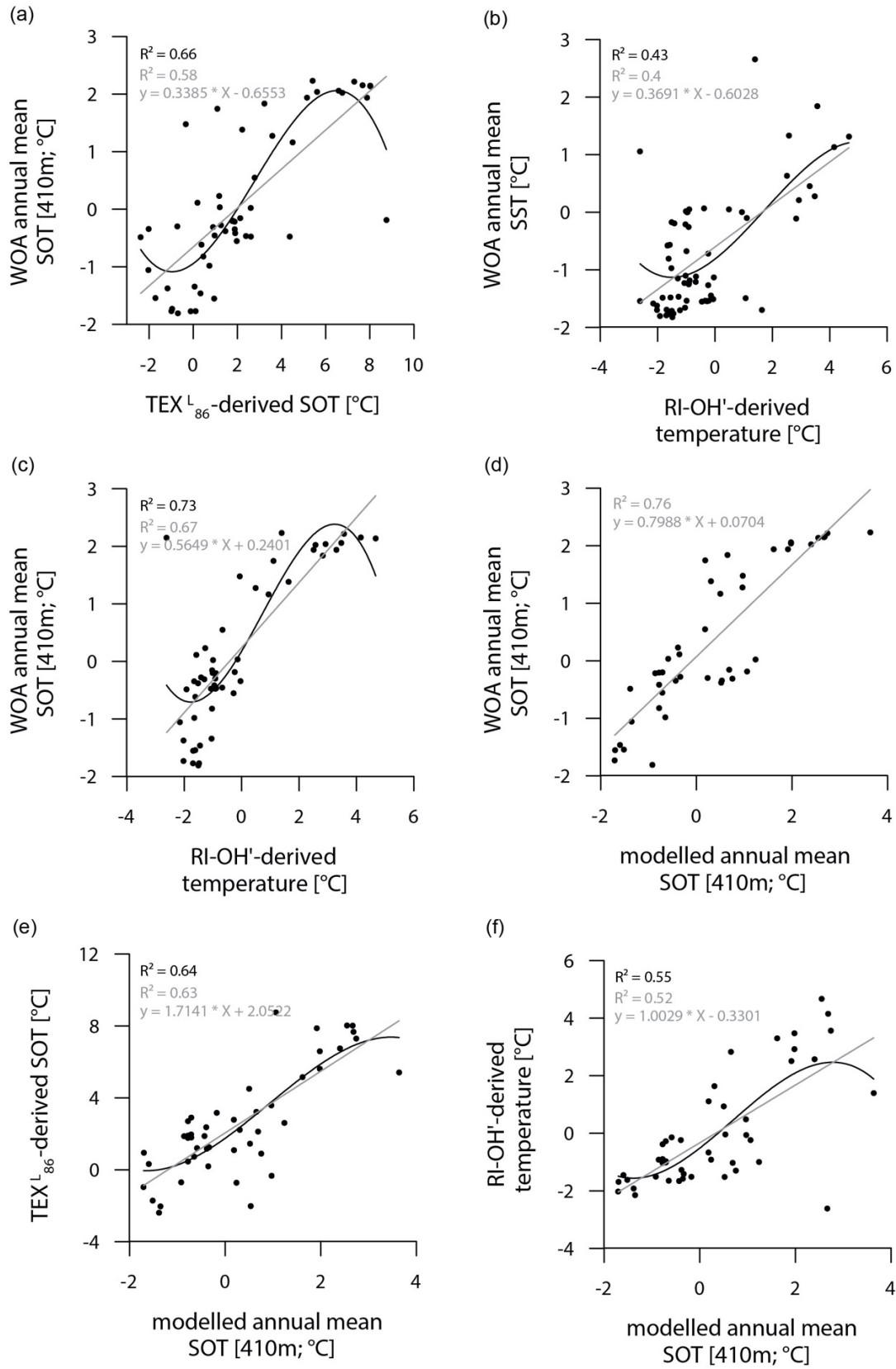

**Fig. 7:** Correlations of (a) WOA annual mean SOT (410 m) vs. TEX$^L_{86}$-derived SOT, (b) WOA annual mean SST vs. RI-OH'-derived temperature, (c) WOA annual mean SOT (410 m) vs. RI-OH'-derived temperature, (d) WOA annual mean SOT (410 m) vs. modelled annual mean SOT (410 m), (e) TEX$^L_{86}$-derived SOT vs. modelled annual mean SOT (410 m), (f) RI-OH'-derived temperature vs. modelled annual mean SOT (410 m) in °C. Coefficients of determination ($R^2$) are given for the respective regression lines.

While the correlation between $TEX^L_{86}$-derived SOTs and instrumental SOTs is reasonably well (Fig.
7a; $R^2 = 0.66$), also supporting the assumption of a subsurface origin, we note a significant
overestimation of SOTs by up to 6 °C in the Drake Passage (Fig. S8). This warm-biased $TEX^L_{86}$ signal
is a known caveat and is, among others, assumed to be connected to GDGTs produced by deep-dwelling
Euryarchaeota (Park et al., 2019), which have been reported in CDW (Alonso-Sáez et al., 2011) and in
deep waters at the Antarctic Polar Front (López-García et al., 2001). Maximum $TEX^L_{86}$-based SOTs of
5 °C - 8 °C in the central Drake Passage (Fig. 6b), however, distinctly exceed the common temperature
range of CDW (0-2 °C). Interestingly, $TEX^L_{86}$-derived SOTs in the colder regions of the Amundsen and
Weddell Seas relate reasonably well to instrumental temperatures and are only slightly warm-biased
(Fig. S8). Correlations between RI-OH'-derived temperatures with instrumental SSTs are
comparatively weak ($R^2 = 0.43$; Fig. 7b). Recently, Liu et al. (2020) concluded in their study on surface
sediments from Prydz Bay (East Antarctica), that also the RI-OH' index holds promise as a tool to
reconstruct SOTs rather than SSTs. When correlating our RI-OH'-derived temperatures with
instrumental SOTs, we find a high correlation ($R^2 = 0.73$; Fig. 7c), too, supporting this hypothesis. We
further note that the temperature range of RI-OH' is much more realistic than $TEX^L_{86}$, supporting the
study by Park et al. (2019) and demonstrating that the addition of OH-isoGDGTs in the temperature
index is a promising step towards high latitude temperature reconstructions and may improve our
understanding of the temperature responses of archaeal membranes in Southern Ocean waters (Fietz et
al., 2020; Park et al., 2019). Clearly, more data – ideally obtained from sediment traps, surface samples
as well as longer sediment cores – and calibration studies will help to further elucidate the applicability
of this approach.
Similar to the model-derived sea-ice data, we here also evaluate the model's performance in depicting
ocean temperatures (Fig. 6e+f). Modelled annual mean SSTs and SOTs are highest with up to 5 °C and
3 °C, respectively, in the permanently ice-free Drake Passage, influenced by the relatively warm ACC.
Decreasing SSTs are simulated towards the Antarctic Peninsula continental slope and the Bransfield
Strait (~0.5 to 1 °C), coinciding with the increase in the duration of seasonal sea-ice cover in that area.
At the EAP/northwestern Weddell Sea, modelled SSTs as well as SOTs show a southwest-northeast
directed increase towards Powell Basin. In the Amundsen and Weddell Seas, annual mean SSTs are
negative, with temperatures ranging from -1 to -0.5 °C, while SOTs are positive in the Amundsen Sea
and negative in the Weddell Sea. Overall, we note that modelled SOTs reflect instrumental SOTs
reasonably well ($R^2$ = 0.76; Fig. 7d). Interestingly, while RI-OH'-derived SOTs relate better to
instrumental SOTs (than $TEX^L_{86}$-based SOTs), we note a better correlation between $TEX^L_{86}$-derived
SOTs and modelled SOTs ($R^2$ = 0.64; Fig. 7e) and a weaker correlation with Ri-OH'-derived
temperatures ($R^2$ = 0.55; Fig. 7f).

**5.  Caveats and recommendations for future research**
Marine core top studies to elucidate the applicability of climate proxies are often concerned with
limitations and uncertainties regarding the age control of the investigated surface sediments as well as
the production, preservation and degradation of target compounds. In the following, we shortly address
some of these factors and provide brief recommendations for future investigations.

5.1  Age control
Information on the actual age of the surface sediments are a major requirement determining their
suitability to reflect modern sea surface conditions. When comparing sea-ice conditions or ocean
temperatures estimated from sedimentary biomarker data (easily spanning decades to millennia,
depending on sedimentation rates) with satellite-derived sea-ice data or instrumental records (covering
only the past ~40 and 65 years, respectively), the different time periods reflected in the data sets need
to be considered when interpreting the results. To address the issue of lacking age constraints for the
herein studied surface sediments, we also performed paleoclimate simulations providing sea-ice
concentration data for three time slices (2 ka, 4 ka and 6 ka BP; see Fig. S5) to evaluate, if the surface
sediments may have recorded significantly older environmental conditions. Correlations of $PIPSO_{25}$
values against these paleo time slice sea-ice concentrations depicted notably weaker relations (Fig. S5)
compared to the recent (1951-2014 CE) model output, which points to a relatively young age of the
majority of the herein studied sediments. This is further supported by AMS [14]C-dating of calcareous
microfossils and $^{210}$Pb-dating of seafloor surface sediments from the Amundsen Sea shelf documenting
recent ages for most sites (Hillenbrand et al., 2010, 2013, 2017; Smith et al., 2011, 2014, 2017; Witus
et al., 2014) as well as modern $^{210}$Pb-dates obtained for three multicores collected in the Bransfield
Strait (PS97/56, PS97/68, PS97/72; Vorrath et al., 2020), which are considered in this study, too. AMS
$^{14}$C dates obtained for nearby surface samples in the vicinity of the South Shetland Islands and the
Antarctic Sound revealed ages of 100 years and 142 years BP, respectively (Vorrath et al., 2019). As
both uncorrected ages lie within the range of the modern marine reservoir effect (e.g. Gordon and
Harkness, 1992), we may consider these two dates still modern. However, in an area that is significantly
affected by rapid climate warming over the past decades and a regionally variable sea ice coverage,
uncertainties associated with $^{14}$C dating of calcareous material may easily lead to an over- or
underestimation of biomarker-based sea-ice cover and ocean temperature estimates, respectively, which
needs to be considered for comparisons with instrumental data. While the utilization of (paleo) model
data may alleviate the lack of age control for each seafloor sediment sample to some extent, we
accordingly recommend that for a robust calibration of e.g. PIPSO$_{25}$ values against satellite-derived sea-
ice concentrations (and this is not the aim of this study) only surface sediment samples with a modern
age confirmed by $^{210}$Pb-dating are incorporated.

5.2    Production and preservation of biomarkers

Biomarkers are considered to reveal the former occurrence of their precursor organisms, which requires
a certain source specificity. While there is general consensus on e.g. Thaumarchaeota being the major
source for iso-GDGTs (Fietz et al., 2020 and references therein) or diatoms synthesizing HBIs
(Volkman 2006), this is not the case for brassicasterol, which is not only found in diatoms but also in
e.g. dinoflagellates and haptophytes (Volkman 2006). Accordingly, the use of brassicasterol to
determine the PIPSO$_{25}$ index may introduce uncertainties regarding the environmental information
pertinent to this phytoplankton biomarker. A further aspect concerns the different chemical structures
of HBIs and sterols, which raises the risk of a selective degradation (see Belt, 2018 and Rontani et al.,
2018; 2019 for detailed discussion) with potentially considerable effects on the PIPSO$_{25}$ index.
Regarding the different sectors of the study area, also spatially different microbial communities as well
as varying depositional regimes, such as sedimentation rate, redox conditions and water depth, may lead
to different degradation patterns, which means that variations in the biomarker concentrations between
different sectors may not strictly reflect changes in the production of these compounds (driven by sea
surface conditions) but may also relate to different degradation states. In particular, lower sedimentation
rates and thus extended oxygen exposure times promote chemical alteration and degradation processes
(Hedges et al., 1990; Schouten et al., 2013). Regarding the transport of organic matter from the sea
surface through the water column, it has been previously noted that the formation of mineral aggregates
and fecal pellets, however, often accelerates the vertical export towards the seafloor during the melting
season leading to a more rapid burial and hence better preservation (Bauerfeind et al., 2005; Etourneau
et al., 2019; Müller et al., 2011).
Another rather technical drawback concerning the use of the $PIPSO_{25}$ index may appear when the
concentrations of the sea-ice proxy $IPSO_{25}$ and the phytoplankton marker are similarly low (due to
unfavourable conditions for both ice algae as well as phytoplankton) or similarly high (due to a
significant seasonal shift in sea-ice cover and/or stable ice edge conditions). This may lead to similar
$PIPSO_{25}$ values, although the sea-ice conditions are fundamentally different from each other. This
scenario occurred at five sampling sites in the Weddell Sea (PS111/13-2, /15-1, /16-3, /29-3, and /40-
2; Fig. 3b+c), where $IPSO_{25}$ and the HBI Z-triene concentrations are close to the detection limit and
$P_ZIPSO_{25}$ values are very low, suggesting a reduced sea-ice cover. Satellite and model data, however,
show that these sample locations are influenced by heavy, nearly year-round sea-ice cover. We conclude
that biomarker concentrations of both biomarkers at or close to the detection limit, indicative of a severe
ice cover, need to be treated with caution. As mentioned above, we assigned a maximum $P_ZIPSO_{25}$
value of 1 to these samples and we note that such practice always needs to be made clear when applying
the $PIPSO_{25}$ approach. The coupling of $IPSO_{25}$ with a phytoplankton marker, nonetheless, provides
more reliable sea-ice reconstructions. Regarding the above-mentioned ambiguities, we recommend not
only to calculate the $PIPSO_{25}$ index, but also to carefully consider individual biomarker concentrations
and, if possible, to utilize other sea-ice measures, such as well-preserved diatom assemblage data
(Lamping et al., 2020; Vorrath et al., 2019; 2020). While the $PIPSO_{25}$ index is not yet a fully quantitative
proxy to provide paleo sea-ice concentrations, the GDGT-paleothermometers have gone through several
calibration iterations (Fietz et al., 2020). As noted above, the observation of distinctly warm-biased
$TEX^L_{86}$-derived SOTs calls for further efforts in terms of regional calibration studies and/or
investigations of archaean adaptation strategies regarding different water depths, nutrient and
temperature conditions.

### 5.3 The role of platelet ice for the production of $IPSO_{25}$
The sympagic, tube-dwelling, diatom *B. adeliensis* is a common constituent of Antarctic sea ice,
preferably flourishing in the relatively open channels of sub-ice platelet layers in near-shore locations
covered by fast ice (Medlin, 1990; Riaux-Gobin and Poulin, 2004). Based on investigations of sea-ice
samples from the Southern Ocean, Belt et al. (2016) detected this diatom species to be a source of
$IPSO_{25}$, which, according to its habitat, led to the assumption of the sea-ice proxy being a potential
indicator for the presence of platelet ice. As stated above, *B. adeliensis* is not confined to platelet ice,
but is also observed in basal sea ice and described as well adapted to changes in the texture of sea ice
during ice melt (Riaux-Gobin et al., 2013). Platelet ice formation, however, plays an important role in
sea-ice generation along some coastal regions of Antarctica (Hoppmann et al., 2015; 2020; Lange et al.,
1989; Langhorne et al., 2015). In these regions, CDW and High Saline Shelf Water (HSSW) flowing
into sub-ice shelf cavities of ice shelves cause basal melting and the discharge of cold and less saline
water (Fig. 8; Hoppmann et al., 2020, Scambos et al., 2017). The surrounding water is cooled and
freshened and is then transported towards the surface. Under the large Filchner-Ronne and Ross ice
shelves the pressure relief can cause this water, called Ice Shelf Water (ISW), to be supercooled (Foldvik
and Kvinge, 1974). The temperature of the supercooled ISW is potentially below the in-situ freezing
point, which may eventually cause the formation of ice platelets that accumulate under landfast ice
attached to adjacent ice shelves (Fig. 8; Holland et al., 2007; Hoppmann et al., 2015; 2020).

**Fig. 8:** Schematic illustration of the formation of platelet ice and the main production areas of sea ice algae producing IPSO$_{25}$ (yellow ovals) and phytoplankton (green ovals), also displayed by yellow and green curves at the top. CDW: Circumpolar Deep Water, HSSW: High Saline Shelf Water, ISW: Ice Shelf Water. Schematic modified after Scambos et al. (2017).

In an attempt to elucidate the relationship of IPSO$_{25}$ and platelet ice more clearly, we here regard our
data in connection to observed platelet ice occurrences.
While the maximum IPSO$_{25}$ concentrations in front of the Filchner Ice Shelf could be directly related
to the above-mentioned platelet ice formation in this area, the elevated IPSO$_{25}$ concentrations in front
of the Larsen Ice Shelves at the EAP could be linked to several processes. According to Langhorne et
al. (2015), sea-ice cores retrieved from that area did not incorporate platelet ice. The high IPSO$_{25}$
concentrations could hence be explicable by either input from drift ice transported with the Weddell
Gyre or by basal freeze-on. We do, however, note that our samples may reflect much longer time frames
than the sea-ice samples investigated by Langhorne et al. (2015) and the lack of platelet ice in their
investigated sea-ice cores does not rule out the former presence of platelet ice, which may be captured
in our investigated sediment samples.
There are several previous studies on IPSO$_{25}$ which report a close connection of the proxy to proximal,
coastal locations and polynyas in the seasonal ice zone (i.e., Collins et al., 2013; Smik et al., 2016).
They do not, however, discuss the relation to adjacent ice shelves as possible "platelet ice factories".
We note that the core locations investigated by Smik et al. (2016) are in the vicinity of the Moscow
University Ice Shelf, where Langhorne et al. (2015) did not observe platelet ice within sea-ice cores.
Hoppmann et al. (2020), however, report a sea-ice core from that area, which incorporates platelet ice.
The different observations by Langhorne et al. (2015) and Hoppmann et al. (2020) highlight the
temporal variability in the occurrence of platelet ice in the cold water regime around the East Antarctic
margin.
Regarding the minimum abundance of $IPSO_{25}$ in the Amundsen Sea (Fig. 3b; AS), which we tentatively
relate to the extended and thick sea ice coverage, the absence of platelet ice in that region may be an
alternative explanation. The Amundsen/Bellingshausen Sea and WAP shelves are classified as warm
shelves (Thompson et al., 2018) characterized by the upwelling of warm CDW (Schmidtko et al., 2014),
hindering the formation of ISW and making the presence of platelet ice in recent conditions highly
unlikely (Hoppmann et al., 2020). This theory is also supported by Langhorne et al. (2015), stating that
platelet ice formation is not observed, where thinning from basal melting of ice shelves is believed to
be greatest, which applies to the warm West Antarctic continental shelf in the eastern Pacific sector of
the Southern Ocean (Thompson et al., 2018). Accordingly, if the formation and accumulation of platelet
ice – up to a certain degree – is indicative of basal ice shelf melting on fresh shelves (Hoppmann et al.,
2015; Thompson et al., 2018), high $IPSO_{25}$ concentrations determined in marine sediments may hence
serve as indicator of ISW formation and associated ice shelf dynamics. This may, however, only be true
up to a certain threshold where platelet ice formation is diminished/hampered due to warm oceanic
conditions causing too intense sub-ice shelf melting (Langhorne et al., 2015).
While using $IPSO_{25}$ as a sea-ice proxy in Antarctica, it is hence important to also consider regional
platelet ice formation processes as these may affect the $IPSO_{25}$ budget. Determining thresholds
associated with platelet ice formation is challenging. Therefore, further investigations, such as in-situ
measurements of $IPSO_{25}$ concentrations in platelet ice or culture experiments in home laboratories, are
needed to better depict the connection between $IPSO_{25}$ and platelet ice formation (and ice shelf basal
melting).

**7.   Conclusions**
Biomarker analyses focusing on $IPSO_{25}$, HBI-trienes, phytosterols and GDGTs in surface sediment
samples from the Antarctic continental margin were investigated to depict recent sea ice conditions and
ocean temperatures in this climate sensitive region. Proxy-based reconstructions of these key variables
were compared to (1) satellite sea-ice data, (2) instrumental ocean temperature data as well as (3)
modelled sea-ice patterns and ocean temperatures. The semi-quantitative sea-ice index $PIPSO_{25}$,
combining the sea-ice proxy $IPSO_{25}$ with an open-water phytoplankton marker, yielded reasonably good
correlations with satellite observations and numerical model results, while correlations with the sea-ice
proxy $IPSO_{25}$ alone are rather low. Minimum concentrations of both biomarkers, used for the $PIPSO_{25}$
calculations, however, may lead to ambiguous interpretations and significant underestimations of sea-
ice conditions. Different sea-ice measures when interpreting biomarker data should hence be
considered.
Ocean temperature reconstructions based on the $TEX^L_{86^-}$ and RI-OH'-paleothermometers show similar
patterns, but different absolute temperatures. While $TEX^L_{86}$-derived temperatures are significantly
biased towards warm temperatures in Drake Passage, the RI-OH'-derived temperature range seems
more realistic, when compared to temperature data based on the WOA13 and modelled annual mean
SOTs.
Further investigations of HBI- as well as GDGT-synthesis, transport, sedimentation and preservation
within the sediments would help to guide the proxies' application. Further taxonomy work, the
composition of the $IPSO_{25}$ producer's habitat (basal sea ice, platelet ice, brine channels) and its
connection to platelet ice formation via in situ or laboratory measurements are required to better
constrain the $IPSO_{25}$ potential as a robust sea-ice biomarker. The presumed relationship between $IPSO_{25}$
and platelet ice formation in connection to basal melting of ice shelves is supported by our data, showing
high $IPSO_{25}$ concentrations in areas where platelet ice formation has previously been reported and low
$IPSO_{25}$ concentrations where no platelet ice formation is observed. Accordingly, oceanic conditions and
the intensity of sub-ice shelf melting need to be considered when using $IPSO_{25}$ (1) as an indirect
indicator for sub-ice shelf melting processes and associated ice shelf dynamics and (2) for the
application of the $PIPSO_{25}$ index to estimate sea ice coverage.

**Data availability**
Datasets related to this article can be found online on *PANGAEA Data Publisher for Earth &*
*Environmental Science* (doi: in prep).

**Author contribution**
N.L. and J.M. designed the concept of the study. N.L. carried out biomarker experiments. X.S and G.L.
developed the model code and X.S. performed the simulations. C.H. provided the satellite data. M.-
E.V. provided hitherto unpublished GDGT data for PS97 samples. G.M. and J.H. carried out GDGT
analyses. C.-D.H. collected surface sediment samples and advised on their ages. N.L. prepared the
manuscript and visualizations with contributions from all co-authors.

**Competing interests**
The authors declare that they have no conflict of interest.

**Acknowledgements**
Denise Diekstall, Mandy Kuck and Jonas Haase are kindly acknowledged for laboratory support. We
thank the captains, crews and science parties of *RV Polarstern* cruises PS69, PS97, PS104, PS111 and
PS118. Especially, Frank Niessen, Sabine Hanisch and Michael Schreck are thanked for their support
during PS118. Simon Belt is acknowledged for providing the 7-HND internal standard for HBI
quantification. AWI, MARUM - University of Bremen, the British Antarctic Survey and NERC UK-
IODP are acknowledged for funding expedition PS104. N.L., M.-E.V. and J.M. were funded through
the Helmholtz Research Grant VH-NG-1101. Two anonymous reviewers are thanked for their
constructive and helpful comments, which lead to a distinct improvement of this manuscript.

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
