# Peer review of "Evaluation of lipid biomarkers as proxies for sea ice and ocean"

_Climate of the Past, 2021_

## Author Comment (AC2)

Supplementary Material

[Figure]

Comparison of ocean temperatures derived from the World Ocean Atlas 13 (WOA13) data base (representing averaged values for the years 1955-2012 (Locarnini et al., 2013); 0.25° resolution) for the sea surface and subsurface (410 m; upper panel), GDGT-based temperatures using RI-OH' and $TEX^L_{86}$ (mid panel) and modelled sea surface and subsurface (410 m) ocean temperatures.

**References**

*Locarnini, R. A., Mishonov, A. V., Antonov, J. I., Boyer, T. P., Garcia, H. E., Baranova, O. K., Zweng, M. M., Paver, C. R., Reagan, J. R., and Johnson, D. R.: World ocean atlas 2013. Volume 1, Temperature, NOAA Atlas NESDIS 73, 40 pp., doi: 10.7289/V55X26VD, 2013.*

[Figure]

[Figure]

[Figure]

Correlations of WOA-derived SSTs against RI-OH'-based temperatures (top left), WOA-derived subsurface (410 m) ocean temperatures with newly calibrated $TEX^L_{86}$-based temperatures (top right), WOA-derived subsurface (410 m) ocean temperatures with RI-OH'-derived temperatures (bottom left).

[Figure]

[Figure]

Correlations of WOA-derived and modelled SSTs (annual mean, 1951-2014; left) and WOA-derived subsurface (410 m) temperatures with modelled subsurface (410 m) ocean temperatures (annual mean, 1951-2014; right).

---

## Author Response (AR1)

**Rebuttal Letter to Editor Comments**

Dear Editor,
thank you very much for your very helpful and constructive comments on our original submission. We considered all your remarks, which - together with the reviewers' suggestions - lead to a significantly improved manuscript. Below, please find our responses to your comments.

Overall request: Please better balance the GDGT and HBI parts, and take extra consideration about the GDGT calibration and subsequent interpretations.
*Reply: We now provide a better-balanced manuscript and, considering a different calibration for subsurface ocean temperatures (SOT), provide and discuss new correlations of GDGT-derived SOTs against instrumental and model SOT data.*

lines 19-20: More recent references?
*Reply: We added Etourneau et al., 2019 and Massom et al., 2018.*

line 27: Consider more recent references: Khazendar et al., NatComms 2016; Smith et al, Nature 2017; Rignot et al., PNAS 2019.
*Reply: We added the above-mentioned articles + Nakayama et al., 2018.*

line 30: How much?
*Reply: We now refer to Fretwell et al. (2013) providing an estimate of 3.4 m (ice grounded on bed below sea level) to 4.3 m (entire WAIS) of global sea level rise, if the WAIS collapses.*

line 45: You rightly mention diatom degradation as a potential bias to past reconstruction, but you do not mention biases to HBIs (Belt, Org Geochem 2018), especially the fact that we do not know how HBIs are transported downward when producers are absent from sediment. An equal evaluation of all proxies is necessary.
*Reply: We rephrased this sentence and added a sub-section (5.2) discussing the roles of biomarker degradation and vertical transport of organic matter.*

lines 103-105: Please present and compare the same metrics in UCDW and LCDW.
*Reply: As we do not further address UCDW and LCDW in the rest of the manuscript we decided to remove this sentence.*

line 122-123 I don't understand if you are speaking about the "mean seasonal cycle" or about decadal changes in the timing of the opening and freezing.
*Reply: We accordingly rephrased this sentence to be more clear about seasonal shifts and decadal changes.*

line 182 (equations): no italic
*Reply: We corrected this.*

lines 214, 216: No italic. Please note that Etourneau et al. NatComms (2019) used a different equation to calculate SST from TEXL86, with smaller constants. Using such an equation will probably reconcile very different TEX-SST and RI-SST as presented in figure 5.
*Reply: As suggested by Reviewer #2 we now use the calibration by Kim et al. (2012) for polar subsurface ocean temperatures (and not SSTs).*

line 325: You said lines 307-309 that IPSO was low in the Powell Basin. Paradoxical to what you say here.
*Reply: We corrected this.*

lines 333-335: In the vicinity of an island, I rather believe that high iron content in water is due to remobilization of shelf sediments along with runoffs. This is observed in the SW Atlantic (de Jong et al., JGR 2012) and around any subantarctic island (Blain et al., Nature 2007; Pollard et al., Nature 2009) where no sea ice is present.
*Reply: We now also refer to the studies by de Jong et al., 2012 and Blain et al., 2007.*

lines 384-385: Any idea why B are not synthesized around SO Island where Triene Z is? I think this is worth commenting as this is the main difference between the two ratios in this region.
*Reply: We now comment on the distinctly lower concentration of brassicasterol in this sample.*

lines 445-448: Deep SO waters never reach values of 10°C, but rather a couple of degrees. So deep-dwelling archea cannot produce such high temperatures or the GDGTs produced by these archea do not fit into the calibration model. If so, a couple of sentences to explain this are necessary. Conversely, this is a question of calibration and you need to use different constants (see comment on the equation).
*Reply: We now discuss the newly calibrated TEXL86-derived SOTs, which are still depicting distinctly higher temperatures in the Drake Passage (compared to World Ocean Atlas[WOA]-derived SOTs) and recommend further calibration efforts (sub-section 5.2).*

lines 450-452: As all plots have different ranges, it is difficult to really compare visually. I think that maps of SST anomalies (WOA - reconstructed SST) would help. At least, in **Supplementary Material.**
*Reply: We changed the temperature ranges accordingly and now provide maps of SOT anomalies as supplementary material.*

line 464: In 4.1.3, reconsructed SST are compared to WOA and modelled SST. I therefore wonder why you did not follow the same scheme for sea ice. You may want to consider merging, and expanding by adding satellite data (fig 2), sections 4.1.2 and 4.1.4.
*Reply: We changed the structure of the results & discussion sections accordingly and now directly consider satellite/WOA and model data for the respective comparison and discussion of proxy-based sea ice and SOT-estimates, respectively.*

lines 567-571: What is the significance of a correlation in which the reconstruction is 7°C higher, with a ~0.5 slope, than the observation? I expect that more statistical work (RMSEP, residuals, t-test...) would show it is not significant.
*Reply: We fully agree that the still significant overestimation of $TEX^L_{86}$-derived SOTs in Drake Passage weakens the reliability of this approach and consequently we now emphasize this issue in the manuscript, also by including WOA-$TEX^L_{86}$- as well as WOA-RI-OH'- temperature anomaly plots in the supplementary material. Furthermore, we recommend in the revised manuscript that more studies focusing on GDGT-synthesis, calibration models and adaptation strategies of archaean need to be conducted to address this problem.*

lines 573-575: How GDGTs produced in CDW of 0-2°C can produce SST as high as 10°C? Any information in Spencer-Jones 2020?
*Reply: We changed this sentence accordingly.*

line 576: Yes, but at least the SST range is valid.
*Reply: We now demonstrate that RI-OH'-derived temperatures seem to reflect SOTs rather than SSTs and highlight the fact that the RI-OH'-derived temperature range is more realistic than that of $TEX^{L}_{86}$-SOTs.*

Figures

Fig. 1: Maximum
*Reply: We corrected this.*

Fig. 5: The figure is misleading as the SST ranges are different in each plot. One may believe that similar colors represent similar SST, which is not the case. Using the same range for all plots might be more sensible, though, I reckon, that RI-OH SST might all fall within blue colors.... Please have a try and chose the best option. Please add core locations (small black dots) on plot (c) to improve visual comparison to other plots.
*Reply: We now use the same temperature range (and colour scale) for all plots and added core locations.*

Fig. 7: I may have missed the information, but why only presenting and discussing PzIPSO? What about $P_{B}$IPSO?
*Reply: We emphasize the better correlation between $P_{Z}IPSO_{25}$ values and satellite data and that we hence focus on $P_{Z}IPSO_{25}$ for the further discussion.*

Fig. 8: If the flyer is not taken into account, a linear regression could be applied. Despite this regression, there is a large offset between reconstructed and observed SST. The linear equation would show constants different than 1 for the slope and 0 for the inception. Indeed, the range of reconstructed SST is twice as large as the range of observed SST.
*Reply: As stated above, we now comment on the warm-bias in $TEX^{L}_{86}$ SOTs. In the revised manuscript, we show both linear and - following Park et al. (2019) - polynomial regressions.*

Fig. 9: I am unsure the HSSW can go up the slope as indicated here. It is formed on the shelf and feel in the sub-ice shelf cavity. I guess this is what you wanted to show but the arrow extends too much over the slope.
*Reply: We corrected this.*

---

## Author Response (AR2)

**Response to Editor Comments**

Dear Editor,
we very much appreciate your constructive comments on our revised manuscript. We considered your remarks/highlighted sections and especially focused on the improvement of the English. Below, you will find our responses to your comments.

line 106: Please check if first time use of sympagic. Definition must be presented at first use.
*Reply: We now present definition at first use (line 38).*

line 176: moieties???
*Reply: A moiety is a functional group of a molecule (OH in this case).*

line 311: Fig 2 does not present decadal changes, but stationary means. Add 1-2 references instead.
*Reply: We now added two references: Vaughan et al. (2003) and Parkinson and Cavalieri (2012).*

line 462: GDGT-3
*Reply: Thanks for spotting this mistake, we corrected it.*

line 550: 2013
*Reply: We corrected it to World Ocean Atlas 2013.*

lines 630-633: Sentence impossible to understand because of too many "and". Please cut in two or rephrase.
*Reply: We changed the sentence to: Highest $IPSO_{25}$ concentrations are observed in samples of the northern Bransfield Strait. Here, the inflow of waters from the Weddell Sea transports sea ice into Bransfield Strait (Vorrath et al., 2019).*

lines 1516-1519: To long and complicated. Cut in two or rephrase.
*Reply: We changed the sentence to: This is also supported by Langhorne et al. (2015), who stated that platelet ice formation is not observed in areas where basal ice-shelf melting is considerable, such as on the West Antarctic continental shelf in the eastern Pacific sector of the Southern Ocean (Thompson et al., 2018).*

line 1651: significant??
*Reply: We replaced "distinct" with "significant".*

References:

Parkinson, C. L., and Cavalieri, D. J.: Antarctic sea ice variability and trends, 1979-2010, The Cryosphere, 6, 871-880, 2012.

Vaughan, D. G., Marshall, G. J., Connolley, W. M., Parkinson, C., Mulvaney, R., Hodgson, D. A., King, J. C., Pudsey, C. J., and Turner, J.: Recent rapid regional climate warming on the Antarctic Peninsula, Climatic change, 60, 243-274, 2003.

---

## Editor Decision (ED2)

[revised manuscript text omitted]

[1] nach oben verschoben: Fig.
[3] nach unten verschoben: 3a; EAP).
[3] verschoben (Einfügung)

[revised manuscript text omitted]

Schriftart: (Standard) Times New Roman

| Seite 1: [9] Formatiert | Nele | 16.08.21 18:50:00 |
|---|---|---|

Schriftart: (Standard) Times New Roman

| Seite 1: [10] Formatiert | Nele | 16.08.21 18:50:00 |
|---|---|---|

Schriftart: (Standard) Times New Roman

| Seite 1: [11] Formatiert | Nele | 16.08.21 18:50:00 |
|---|---|---|

Block, Keine, Zeilenabstand:  Doppelt

| Seite 1: [12] Formatiert | Nele | 16.08.21 18:50:00 |
|---|---|---|

Schriftart: (Standard) Times New Roman, Nicht Hochgestellt/ Tiefgestellt

| Seite 1: [13] Formatiert | Nele | 16.08.21 18:50:00 |
|---|---|---|

Schriftart: (Standard) Times New Roman

| Seite 1: [14] Formatiert | Nele | 16.08.21 18:50:00 |
|---|---|---|

Schriftart: (Standard) Times New Roman, Hochgestellt

| Seite 1: [15] Formatiert | Nele | 16.08.21 18:50:00 |
|---|---|---|

Schriftart: (Standard) Times New Roman, 11 Pt.

| Seite 1: [16] Formatiert | Nele | 16.08.21 18:50:00 |
|---|---|---|

Block, Zeilenabstand:  Doppelt

| Seite 1: [17] Formatiert | Nele | 16.08.21 18:50:00 |
|---|---|---|

Schriftart: (Standard) Times New Roman, 11 Pt.

| Seite 1: [18] Formatiert | Nele | 16.08.21 18:50:00 |
|---|---|---|

Schriftart: (Standard) Times New Roman, 11 Pt.

| Seite 1: [19] Formatiert | Nele | 16.08.21 18:50:00 |
|---|---|---|

Schriftart: (Standard) Times New Roman, 11 Pt.

| Seite 1: [20] Formatiert | Nele | 16.08.21 18:50:00 |
|---|---|---|

Schriftart: (Standard) Times New Roman, 11 Pt.

| Seite 1: [21] Formatiert | Nele | 16.08.21 18:50:00 |
|---|---|---|

Schriftart: (Standard) Times New Roman, 11 Pt., Englisch (Vereinigtes Königreich)

| Seite 1: [22] Gelöscht | Nele | 16.08.21 18:50:00 |
|---|---|---|

| Seite 1: [23] Formatiert | Nele | 16.08.21 18:50:00 |
|---|---|---|

Schriftart: (Standard) Times New Roman, 11 Pt.

| Seite 1: [24] Formatiert | Nele | 16.08.21 18:50:00 |
|---|---|---|

Block, Zeilenabstand:  Doppelt

| Seite 1: [25] Formatiert | Nele | 16.08.21 18:50:00 |
|---|---|---|

Schriftart: (Standard) Times New Roman

| Seite 1: [26] Formatiert | Nele | 16.08.21 18:50:00 |
|---|---|---|

Block, Zeilenabstand:  Doppelt, Rahmen: Unten: (Kein Rahmen)

| Seite 1: [27] Formatiert | Nele | 16.08.21 18:50:00 |
|---|---|---|

Schriftart: (Standard) Times New Roman, Kursiv

| Seite 1: [28] Formatiert | Nele | 16.08.21 18:50:00 |
|---|---|---|

Block, Zeilenabstand:  Doppelt

| Seite 1: [29] Formatiert | Nele | 16.08.21 18:50:00 |
|---|---|---|

Schriftart: (Standard) Times New Roman

| Seite 1: [30] Formatiert | Nele | 16.08.21 18:50:00 |
|---|---|---|

Schriftart: (Standard) Times New Roman

| Seite 1: [31] Formatiert | Nele | 16.08.21 18:50:00 |
|---|---|---|

Schriftart: (Standard) Times New Roman

| Seite 1: [31] Formatiert | Nele | 16.08.21 18:50:00 |
|---|---|---|

Schriftart: (Standard) Times New Roman

| Seite 1: [32] Formatiert | Nele | 16.08.21 18:50:00 |
|---|---|---|

Schriftart: (Standard) Times New Roman

| Seite 1: [32] Formatiert | Nele | 16.08.21 18:50:00 |
|---|---|---|

Schriftart: (Standard) Times New Roman

| Seite 1: [32] Formatiert | Nele | 16.08.21 18:50:00 |
|---|---|---|

Schriftart: (Standard) Times New Roman

| Seite 1: [33] Formatiert | Nele | 16.08.21 18:50:00 |
|---|---|---|

Schriftart: (Standard) Times New Roman

| Seite 1: [34] Formatiert | Nele | 16.08.21 18:50:00 |
|---|---|---|

Schriftart: (Standard) Times New Roman

| Seite 1: [35] Formatiert | Nele | 16.08.21 18:50:00 |
|---|---|---|

Schriftart: (Standard) Times New Roman

| Seite 1: [36] Formatiert | Nele | 16.08.21 18:50:00 |
|---|---|---|

Schriftart: (Standard) Times New Roman

| Seite 1: [37] Formatiert | Nele | 16.08.21 18:50:00 |
|---|---|---|

Schriftart: (Standard) Times New Roman

| Seite 1: [38] Formatiert | Nele | 16.08.21 18:50:00 |
|---|---|---|

Schriftart: (Standard) Times New Roman

| Seite 1: [39] Formatiert | Nele | 16.08.21 18:50:00 |
|---|---|---|

Schriftart: (Standard) Times New Roman

| Seite 1: [40] Formatiert | Nele | 16.08.21 18:50:00 |
|---|---|---|

Schriftart: (Standard) Times New Roman

| Seite 1: [40] Formatiert | Nele | 16.08.21 18:50:00 |
|---|---|---|

Schriftart: (Standard) Times New Roman

| Seite 1: [41] Formatiert | Nele | 16.08.21 18:50:00 |
|---|---|---|

Schriftart: (Standard) Times New Roman

| Seite 1: [42] Formatiert | Nele | 16.08.21 18:50:00 |
|---|---|---|

Schriftart: (Standard) Times New Roman

| Seite 1: [43] Formatiert | Nele | 16.08.21 18:50:00 |
|---|---|---|

Schriftart: (Standard) Times New Roman

| Seite 32: [44] Formatiert | Nele | 16.08.21 18:50:00 |
|---|---|---|

Nach:  0.63 cm

| Seite 1: [45] Formatiert | Nele | 16.08.21 18:50:00 |
|---|---|---|

Standard, Block, Einzug: Vor:  0.74 cm, Hängend:  0.74 cm, Zeilenabstand:  Doppelt,  Keine Aufzählungen oder Nummerierungen

| Seite 1: [46] Formatiert | Nele | 16.08.21 18:50:00 |
|---|---|---|

Schriftart: (Standard) Times New Roman

| Seite 1: [47] Formatiert | Nele | 16.08.21 18:50:00 |
|---|---|---|

Block, Zeilenabstand:  Doppelt

| Seite 1: [48] Formatiert | Nele | 16.08.21 18:50:00 |
|---|---|---|

Schriftart: (Standard) Times New Roman, Schriftfarbe: Automatisch

| Seite 1: [49] Formatiert | Nele | 16.08.21 18:50:00 |
|---|---|---|

Schriftart: (Standard) Times New Roman, Schriftfarbe: Automatisch

| Seite 1: [49] Formatiert | Nele | 16.08.21 18:50:00 |
|---|---|---|

Schriftart: (Standard) Times New Roman, Schriftfarbe: Automatisch

| Seite 1: [50] Formatiert | Nele | 16.08.21 18:50:00 |
|---|---|---|

Schriftart: (Standard) Times New Roman

| Seite 1: [51] Formatiert | Nele | 16.08.21 18:50:00 |
|---|---|---|

Schriftart: (Standard) Times New Roman, Hervorheben

| Seite 1: [52] Gelöscht | Nele | 16.08.21 18:50:00 |
|---|---|---|

| Seite 1: [53] Formatiert | Nele | 16.08.21 18:50:00 |
|---|---|---|

Schriftart: (Standard) Times New Roman

| Seite 1: [53] Formatiert | Nele | 16.08.21 18:50:00 |
|---|---|---|

Schriftart: (Standard) Times New Roman

| Seite 1: [54] Formatiert | Nele | 16.08.21 18:50:00 |
|---|---|---|

Schriftart: (Standard) Times New Roman

| Seite 1: [55] Formatiert | Nele | 16.08.21 18:50:00 |
|---|---|---|

Schriftart: (Standard) Times New Roman

| Seite 1: [56] Gelöscht | Nele | 16.08.21 18:50:00 |
|---|---|---|

| Seite 1: [57] Formatiert | Nele | 16.08.21 18:50:00 |
|---|---|---|

Schriftart: (Standard) Times New Roman

| Seite 1: [58] Formatiert | Nele | 16.08.21 18:50:00 |
|---|---|---|

Schriftart: (Standard) Times New Roman

| Seite 1: [59] Gelöscht | Nele | 16.08.21 18:50:00 |
|---|---|---|

| Seite 1: [60] Formatiert | Nele | 16.08.21 18:50:00 |
|---|---|---|

Schriftart: (Standard) Times New Roman, Schriftfarbe: Automatisch

| Seite 1: [61] Formatiert | Nele | 16.08.21 18:50:00 |
|---|---|---|

Schriftart: (Standard) Times New Roman

| Seite 1: [62] Formatiert | Nele | 16.08.21 18:50:00 |
|---|---|---|

Schriftart: (Standard) Times New Roman

| Seite 1: [62] Formatiert | Nele | 16.08.21 18:50:00 |
|---|---|---|

Schriftart: (Standard) Times New Roman

| Seite 1: [63] Formatiert | Nele | 16.08.21 18:50:00 |
|---|---|---|

Schriftart: (Standard) Times New Roman

| Seite 1: [64] Gelöscht | Nele | 16.08.21 18:50:00 |
|---|---|---|

| Seite 1: [65] Formatiert | Nele | 16.08.21 18:50:00 |
|---|---|---|

Schriftart: (Standard) Times New Roman

| Seite 1: [66] Formatiert | Nele | 16.08.21 18:50:00 |
|---|---|---|

Schriftart: (Standard) Times New Roman

| Seite 1: [67] Formatiert | Nele | 16.08.21 18:50:00 |
|---|---|---|

Schriftart: (Standard) Times New Roman

| Seite 1: [68] Formatiert | Nele | 16.08.21 18:50:00 |
|---|---|---|

Schriftart: (Standard) Times New Roman

| Seite 1: [69] Formatiert | Nele | 16.08.21 18:50:00 |
|---|---|---|

Schriftart: (Standard) Times New Roman

| Seite 1: [70] Formatiert | Nele | 16.08.21 18:50:00 |
|---|---|---|

Schriftart: (Standard) Times New Roman

| Seite 1: [71] Formatiert | Nele | 16.08.21 18:50:00 |
|---|---|---|

Schriftart: (Standard) Times New Roman

| Seite 1: [71] Formatiert | Nele | 16.08.21 18:50:00 |
|---|---|---|

Schriftart: (Standard) Times New Roman

| Seite 1: [72] Formatiert | Nele | 16.08.21 18:50:00 |
|---|---|---|

Schriftart: (Standard) Times New Roman

| Seite 1: [73] Formatiert | Nele | 16.08.21 18:50:00 |
|---|---|---|

Schriftart: (Standard) Times New Roman

| Seite 1: [74] Formatiert | Nele | 16.08.21 18:50:00 |
|---|---|---|

Schriftart: (Standard) Times New Roman

| Seite 1: [74] Formatiert | Nele | 16.08.21 18:50:00 |
|---|---|---|

Schriftart: (Standard) Times New Roman

| Seite 1: [74] Formatiert | Nele | 16.08.21 18:50:00 |
|---|---|---|

Schriftart: (Standard) Times New Roman

| Seite 1: [75] Formatiert | Nele | 16.08.21 18:50:00 |
|---|---|---|

Schriftart: (Standard) Times New Roman

| Seite 1: [76] Formatiert | Nele | 16.08.21 18:50:00 |
|---|---|---|

Schriftart: (Standard) Times New Roman

| Seite 1: [77] Formatiert | Nele | 16.08.21 18:50:00 |
|---|---|---|

Schriftart: (Standard) Times New Roman

| Seite 1: [77] Formatiert | Nele | 16.08.21 18:50:00 |
|---|---|---|

Schriftart: (Standard) Times New Roman

| Seite 2: [78] Formatiert | Nele | 16.08.21 18:50:00 |
|---|---|---|

Schriftart: (Standard) Times New Roman, Schriftfarbe: Automatisch

| Seite 2: [79] Formatiert | Nele | 16.08.21 18:50:00 |
|---|---|---|

Schriftart: (Standard) Times New Roman, Schriftfarbe: Automatisch

| Seite 2: [80] Formatiert | Nele | 16.08.21 18:50:00 |
|---|---|---|

Schriftart: (Standard) Times New Roman, Schriftfarbe: Automatisch

| Seite 2: [81] Formatiert | Nele | 16.08.21 18:50:00 |
|---|---|---|

Schriftart: (Standard) Times New Roman, Schriftfarbe: Automatisch

| Seite 2: [82] Formatiert | Nele | 16.08.21 18:50:00 |
|---|---|---|

Schriftart: (Standard) Times New Roman, Schriftfarbe: Automatisch

| Seite 2: [83] Formatiert | Nele | 16.08.21 18:50:00 |
|---|---|---|

Schriftart: (Standard) Times New Roman, Schriftfarbe: Automatisch

| Seite 2: [84] Formatiert | Nele | 16.08.21 18:50:00 |
|---|---|---|

Schriftart: (Standard) Times New Roman, Kursiv, Schriftfarbe: Automatisch

| Seite 2: [85] Formatiert | Nele | 16.08.21 18:50:00 |
|---|---|---|

Schriftart: (Standard) Times New Roman, Schriftfarbe: Automatisch

| Seite 2: [86] Formatiert | Nele | 16.08.21 18:50:00 |
|---|---|---|

Schriftart: (Standard) Times New Roman, Schriftfarbe: Automatisch

| Seite 2: [87] Gelöscht | Nele | 16.08.21 18:50:00 |
|---|---|---|

| Seite 2: [88] Formatiert | Nele | 16.08.21 18:50:00 |
|---|---|---|

Schriftart: (Standard) Times New Roman, Schriftfarbe: Automatisch

| Seite 2: [89] Formatiert | Nele | 16.08.21 18:50:00 |
|---|---|---|

Schriftart: (Standard) Times New Roman, Schriftfarbe: Automatisch

| Seite 2: [90] Formatiert | Nele | 16.08.21 18:50:00 |
|---|---|---|

Schriftart: (Standard) Times New Roman, Schriftfarbe: Automatisch

| Seite 2: [91] Formatiert | Nele | 16.08.21 18:50:00 |
|---|---|---|

Schriftart: (Standard) Times New Roman, Schriftfarbe: Automatisch

| Seite 32: [92] Formatiert | Nele | 16.08.21 18:50:00 |
|---|---|---|

Nach:  0.63 cm

| Seite 3: [93] Formatiert | Nele | 16.08.21 18:50:00 |
| --- | --- | --- |

Schriftart: (Standard) Times New Roman, Schriftfarbe: Automatisch

| Seite 3: [94] Gelöscht | Nele | 16.08.21 18:50:00 |
| --- | --- | --- |

▼

| Seite 3: [95] Formatiert | Nele | 16.08.21 18:50:00 |
| --- | --- | --- |

Schriftart: (Standard) Times New Roman, Schriftfarbe: Automatisch

| Seite 3: [96] Formatiert | Nele | 16.08.21 18:50:00 |
| --- | --- | --- |

Schriftart: (Standard) Times New Roman, Schriftfarbe: Automatisch

| Seite 3: [97] Formatiert | Nele | 16.08.21 18:50:00 |
| --- | --- | --- |

Schriftart: (Standard) Times New Roman, Schriftfarbe: Automatisch

| Seite 3: [98] Formatiert | Nele | 16.08.21 18:50:00 |
| --- | --- | --- |

Schriftart: (Standard) Times New Roman, Schriftfarbe: Automatisch

| Seite 3: [99] Formatiert | Nele | 16.08.21 18:50:00 |
| --- | --- | --- |

Schriftart: (Standard) Times New Roman, Schriftfarbe: Automatisch

| Seite 3: [100] Formatiert | Nele | 16.08.21 18:50:00 |
| --- | --- | --- |

Schriftart: (Standard) Times New Roman, Schriftfarbe: Automatisch

| Seite 3: [101] Formatiert | Nele | 16.08.21 18:50:00 |
| --- | --- | --- |

Schriftart: (Standard) Times New Roman, Schriftfarbe: Automatisch

| Seite 3: [102] Formatiert | Nele | 16.08.21 18:50:00 |
| --- | --- | --- |

Schriftart: (Standard) Times New Roman, Schriftfarbe: Automatisch

| Seite 3: [103] Formatiert | Nele | 16.08.21 18:50:00 |
| --- | --- | --- |

Schriftart: (Standard) Times New Roman, Schriftfarbe: Automatisch

| Seite 3: [104] Formatiert | Nele | 16.08.21 18:50:00 |
| --- | --- | --- |

Schriftart: (Standard) Times New Roman, Schriftfarbe: Automatisch

| Seite 3: [104] Formatiert | Nele | 16.08.21 18:50:00 |
| --- | --- | --- |

Schriftart: (Standard) Times New Roman, Schriftfarbe: Automatisch

| Seite 3: [105] Formatiert | Nele | 16.08.21 18:50:00 |
| --- | --- | --- |

Schriftart: (Standard) Times New Roman, Schriftfarbe: Automatisch

| Seite 3: [106] Formatiert | Nele | 16.08.21 18:50:00 |
| --- | --- | --- |

Schriftart: (Standard) Times New Roman, Schriftfarbe: Automatisch

| Seite 3: [107] Formatiert | Nele | 16.08.21 18:50:00 |
| --- | --- | --- |

Schriftart: (Standard) Times New Roman, Schriftfarbe: Automatisch

| Seite 3: [108] Formatiert | Nele | 16.08.21 18:50:00 |
| --- | --- | --- |

Schriftart: (Standard) Times New Roman, Schriftfarbe: Automatisch

| Seite 3: [109] Formatiert | Nele | 16.08.21 18:50:00 |
| --- | --- | --- |

Schriftart: (Standard) Times New Roman, Schriftfarbe: Automatisch

| Seite 3: [110] Formatiert | Nele | 16.08.21 18:50:00 |
| --- | --- | --- |

Schriftart: (Standard) Times New Roman, Schriftfarbe: Automatisch

| Seite 3: [110] Formatiert | Nele | 16.08.21 18:50:00 |
|---|---|---|

Schriftart: (Standard) Times New Roman, Schriftfarbe: Automatisch

| Seite 3: [111] Formatiert | Nele | 16.08.21 18:50:00 |
|---|---|---|

Schriftart: (Standard) Times New Roman, Schriftfarbe: Automatisch

| Seite 3: [112] Formatiert | Nele | 16.08.21 18:50:00 |
|---|---|---|

Schriftart: (Standard) Times New Roman, Schriftfarbe: Automatisch

| Seite 3: [113] Formatiert | Nele | 16.08.21 18:50:00 |
|---|---|---|

Schriftart: (Standard) Times New Roman, Schriftfarbe: Automatisch

| Seite 3: [113] Formatiert | Nele | 16.08.21 18:50:00 |
|---|---|---|

Schriftart: (Standard) Times New Roman, Schriftfarbe: Automatisch

| Seite 3: [114] Formatiert | Nele | 16.08.21 18:50:00 |
|---|---|---|

Schriftart: (Standard) Times New Roman, Schriftfarbe: Automatisch

| Seite 3: [115] Formatiert | Nele | 16.08.21 18:50:00 |
|---|---|---|

Schriftart: (Standard) Times New Roman, Schriftfarbe: Automatisch

| Seite 3: [115] Formatiert | Nele | 16.08.21 18:50:00 |
|---|---|---|

Schriftart: (Standard) Times New Roman, Schriftfarbe: Automatisch

| Seite 3: [115] Formatiert | Nele | 16.08.21 18:50:00 |
|---|---|---|

Schriftart: (Standard) Times New Roman, Schriftfarbe: Automatisch

| Seite 3: [116] Gelöscht | Nele | 16.08.21 18:50:00 |
|---|---|---|

| Seite 32: [117] Formatiert | Nele | 16.08.21 18:50:00 |
|---|---|---|

Nach:  0.63 cm

| Seite 7: [118] Formatiert | Nele | 16.08.21 18:50:00 |
|---|---|---|

Schriftart: (Standard) Times New Roman, Englisch (Vereinigtes Königreich)

| Seite 7: [119] Formatiert | Nele | 16.08.21 18:50:00 |
|---|---|---|

Block, Einzug: Vor:  0.74 cm, Hängend:  0.74 cm, Zeilenabstand:  Doppelt

| Seite 7: [120] Formatiert | Nele | 16.08.21 18:50:00 |
|---|---|---|

Schriftart: (Standard) Times New Roman, Nicht Fett, Englisch (Vereinigtes Königreich)

| Seite 7: [121] Formatiert | Nele | 16.08.21 18:50:00 |
|---|---|---|

Schriftart: (Standard) Times New Roman

| Seite 7: [122] Formatiert | Nele | 16.08.21 18:50:00 |
|---|---|---|

Schriftart: (Standard) Times New Roman

| Seite 7: [123] Formatiert | Nele | 16.08.21 18:50:00 |
|---|---|---|

Block, Zeilenabstand:  Doppelt

| Seite 7: [124] Formatiert | Nele | 16.08.21 18:50:00 |
|---|---|---|

Schriftart: (Standard) Times New Roman

| Seite 7: [124] Formatiert | Nele | 16.08.21 18:50:00 |
|---|---|---|

Schriftart: (Standard) Times New Roman

| Seite 7: [125] Formatiert | Nele | 16.08.21 18:50:00 |
|---|---|---|

Schriftart: (Standard) Times New Roman

| Seite 7: [125] Formatiert | Nele | 16.08.21 18:50:00 |

Schriftart: (Standard) Times New Roman

| Seite 7: [126] Formatiert | Nele | 16.08.21 18:50:00 |

Schriftart: (Standard) Times New Roman

| Seite 7: [127] Formatiert | Nele | 16.08.21 18:50:00 |

Schriftart: (Standard) Times New Roman

| Seite 7: [128] Formatiert | Nele | 16.08.21 18:50:00 |

Schriftart: (Standard) Times New Roman

| Seite 7: [129] Formatiert | Nele | 16.08.21 18:50:00 |

Schriftart: (Standard) Times New Roman

| Seite 7: [130] Formatiert | Nele | 16.08.21 18:50:00 |

Schriftart: (Standard) Times New Roman

| Seite 7: [131] Formatiert | Nele | 16.08.21 18:50:00 |

Schriftart: (Standard) Times New Roman

| Seite 7: [132] Formatiert | Nele | 16.08.21 18:50:00 |

Standard, Block, Einzug: Vor:  1 cm, Hängend:  1 cm, Zeilenabstand:  Doppelt,  Keine Aufzählungen oder Nummerierungen

| Seite 7: [133] Formatiert | Nele | 16.08.21 18:50:00 |

Schriftart: (Standard) Times New Roman

| Seite 7: [134] Formatiert | Nele | 16.08.21 18:50:00 |

Block, Zeilenabstand:  Doppelt

| Seite 7: [135] Formatiert | Nele | 16.08.21 18:50:00 |

Schriftart: (Standard) Times New Roman

| Seite 7: [136] Formatiert | Nele | 16.08.21 18:50:00 |

Schriftart: (Standard) Times New Roman

| Seite 7: [137] Formatiert | Nele | 16.08.21 18:50:00 |

Schriftart: (Standard) Times New Roman

| Seite 7: [138] Formatiert | Nele | 16.08.21 18:50:00 |

Schriftart: (Standard) Times New Roman

| Seite 7: [139] Formatiert | Nele | 16.08.21 18:50:00 |

Schriftart: (Standard) Times New Roman

| Seite 7: [139] Formatiert | Nele | 16.08.21 18:50:00 |

Schriftart: (Standard) Times New Roman

| Seite 7: [140] Formatiert | Nele | 16.08.21 18:50:00 |

Schriftart: (Standard) +Überschriften CS (Times New Roman)

| Seite 7: [141] Formatiert | Nele | 16.08.21 18:50:00 |

Schriftart: (Standard) +Überschriften CS (Times New Roman)

| Seite 7: [141] Formatiert | Nele | 16.08.21 18:50:00 |

Schriftart: (Standard) +Überschriften CS (Times New Roman)

| Seite 7: [142] Formatiert | Nele | 16.08.21 18:50:00 |
|---|---|---|

Schriftart: (Standard) +Überschriften CS (Times New Roman)

| Seite 7: [143] Formatiert | Nele | 16.08.21 18:50:00 |
|---|---|---|

Schriftart: (Standard) +Überschriften CS (Times New Roman)

| Seite 7: [143] Formatiert | Nele | 16.08.21 18:50:00 |
|---|---|---|

Schriftart: (Standard) +Überschriften CS (Times New Roman)

| Seite 32: [144] Formatiert | Nele | 16.08.21 18:50:00 |
|---|---|---|

Nach: 0.63 cm

| Seite 12: [145] Formatiert | Nele | 16.08.21 18:50:00 |
|---|---|---|

Schriftart: (Standard) Times New Roman, Englisch (Vereinigtes Königreich)

| Seite 12: [146] Formatiert | Nele | 16.08.21 18:50:00 |
|---|---|---|

Schriftart: (Standard) +Überschriften CS (Times New Roman), 10 Pt., Fett, Schriftfarbe: Text 1, Englisch (Vereinigtes Königreich)

| Seite 12: [147] Formatiert | Nele | 16.08.21 18:50:00 |
|---|---|---|

Schriftart: (Standard) Times New Roman, Englisch (Vereinigtes Königreich)

| Seite 12: [148] Formatiert | Nele | 16.08.21 18:50:00 |
|---|---|---|

Standard, Block, Einzug: Vor: 0.74 cm, Hängend: 0.74 cm, Zeilenabstand: Doppelt, Keine Aufzählungen oder Nummerierungen

| Seite 12: [149] Formatiert | Nele | 16.08.21 18:50:00 |
|---|---|---|

Schriftart: (Standard) Times New Roman

| Seite 12: [150] Gelöscht | Nele | 16.08.21 18:50:00 |
|---|---|---|

| Seite 12: [151] Formatiert | Nele | 16.08.21 18:50:00 |
|---|---|---|

Block, Zeilenabstand: Doppelt

| Seite 12: [152] Formatiert | Nele | 16.08.21 18:50:00 |
|---|---|---|

Schriftart: (Standard) Times New Roman, Schriftfarbe: Automatisch

| Seite 12: [153] Formatiert | Nele | 16.08.21 18:50:00 |
|---|---|---|

Schriftart: (Standard) Times New Roman, Schriftfarbe: Automatisch

| Seite 12: [154] Formatiert | Nele | 16.08.21 18:50:00 |
|---|---|---|

Schriftart: (Standard) Times New Roman, Schriftfarbe: Automatisch

| Seite 12: [154] Formatiert | Nele | 16.08.21 18:50:00 |
|---|---|---|

Schriftart: (Standard) Times New Roman, Schriftfarbe: Automatisch

| Seite 12: [154] Formatiert | Nele | 16.08.21 18:50:00 |
|---|---|---|

Schriftart: (Standard) Times New Roman, Schriftfarbe: Automatisch

| Seite 12: [154] Formatiert | Nele | 16.08.21 18:50:00 |
|---|---|---|

Schriftart: (Standard) Times New Roman, Schriftfarbe: Automatisch

| Seite 12: [154] Formatiert | Nele | 16.08.21 18:50:00 |
|---|---|---|

Schriftart: (Standard) Times New Roman, Schriftfarbe: Automatisch

| Seite 12: [155] Formatiert | Nele | 16.08.21 18:50:00 |
|---|---|---|

Standard, Block, Einzug: Vor: 1 cm, Hängend: 1 cm, Zeilenabstand: Doppelt, Keine Aufzählungen oder Nummerierungen

| Seite 12: [156] Formatiert | Nele | 16.08.21 18:50:00 |
|---|---|---|

Schriftart: (Standard) Times New Roman

| Seite 12: [157] Formatiert | Nele | 16.08.21 18:50:00 |
|---|---|---|

Schriftart: (Standard) Times New Roman, 9 Pt.

| Seite 12: [158] Formatiert | Nele | 16.08.21 18:50:00 |
|---|---|---|

Block, Zeilenabstand:  Doppelt

| Seite 12: [159] Formatiert | Nele | 16.08.21 18:50:00 |
|---|---|---|

Schriftart: (Standard) Times New Roman

| Seite 12: [160] Formatiert | Nele | 16.08.21 18:50:00 |
|---|---|---|

Schriftart: (Standard) Times New Roman

| Seite 12: [161] Formatiert | Nele | 16.08.21 18:50:00 |
|---|---|---|

Schriftart: (Standard) Times New Roman

| Seite 12: [162] Formatiert | Nele | 16.08.21 18:50:00 |
|---|---|---|

Schriftart: (Standard) Times New Roman

| Seite 12: [163] Formatiert | Nele | 16.08.21 18:50:00 |
|---|---|---|

Schriftart: (Standard) Times New Roman

| Seite 12: [164] Formatiert | Nele | 16.08.21 18:50:00 |
|---|---|---|

Schriftart: (Standard) Times New Roman

| Seite 12: [165] Formatiert | Nele | 16.08.21 18:50:00 |
|---|---|---|

Schriftart: (Standard) Times New Roman

| Seite 12: [166] Formatiert | Nele | 16.08.21 18:50:00 |
|---|---|---|

Schriftart: (Standard) Times New Roman

| Seite 12: [167] Formatiert | Nele | 16.08.21 18:50:00 |
|---|---|---|

Schriftart: (Standard) Times New Roman

| Seite 12: [168] Gelöscht | Nele | 16.08.21 18:50:00 |
|---|---|---|

| Seite 12: [169] Formatiert | Nele | 16.08.21 18:50:00 |
|---|---|---|

Schriftart: (Standard) Times New Roman

| Seite 12: [170] Formatiert | Nele | 16.08.21 18:50:00 |
|---|---|---|

Schriftart: (Standard) Times New Roman

| Seite 12: [171] Formatiert | Nele | 16.08.21 18:50:00 |
|---|---|---|

Schriftart: (Standard) Times New Roman

| Seite 14: [172] Formatiert | Nele | 16.08.21 18:50:00 |
|---|---|---|

Schriftart: (Standard) Times New Roman, Schriftfarbe: Automatisch

| Seite 14: [173] Formatiert | Nele | 16.08.21 18:50:00 |
|---|---|---|

Schriftart: (Standard) Times New Roman, Schriftfarbe: Automatisch

| Seite 14: [174] Formatiert | Nele | 16.08.21 18:50:00 |
|---|---|---|

Schriftart: (Standard) Times New Roman, Schriftfarbe: Automatisch

| Seite 32: [175] Formatiert | Nele | 16.08.21 18:50:00 |
|---|---|---|

Nach: 0.63 cm

| Seite 15: [176] Formatiert | Nele | 16.08.21 18:50:00 |
|---|---|---|

Schriftart: (Standard) Times New Roman

| Seite 15: [177] Formatiert | Nele | 16.08.21 18:50:00 |
|---|---|---|

Schriftart: (Standard) Times New Roman

| Seite 15: [178] Gelöscht | Nele | 16.08.21 18:50:00 |
|---|---|---|

| Seite 15: [179] Formatiert | Nele | 16.08.21 18:50:00 |
|---|---|---|

Schriftart: (Standard) Times New Roman

| Seite 15: [180] Formatiert | Nele | 16.08.21 18:50:00 |
|---|---|---|

Schriftart: (Standard) Times New Roman

| Seite 15: [181] Formatiert | Nele | 16.08.21 18:50:00 |
|---|---|---|

Schriftart: (Standard) Times New Roman

| Seite 15: [182] Formatiert | Nele | 16.08.21 18:50:00 |
|---|---|---|

Schriftart: (Standard) Times New Roman

| Seite 15: [182] Formatiert | Nele | 16.08.21 18:50:00 |
|---|---|---|

Schriftart: (Standard) Times New Roman

| Seite 15: [183] Formatiert | Nele | 16.08.21 18:50:00 |
|---|---|---|

Schriftart: (Standard) Times New Roman

| Seite 15: [184] Gelöscht | Nele | 16.08.21 18:50:00 |
|---|---|---|

| Seite 15: [185] Formatiert | Nele | 16.08.21 18:50:00 |
|---|---|---|

Schriftart: (Standard) +Überschriften CS (Times New Roman), 10 Pt., Fett, Schriftfarbe: Text 1

| Seite 15: [186] Gelöscht | Nele | 16.08.21 18:50:00 |
|---|---|---|

| Seite 15: [187] Formatiert | Nele | 16.08.21 18:50:00 |
|---|---|---|

Schriftart: (Standard) Times New Roman

| Seite 15: [188] Formatiert | Nele | 16.08.21 18:50:00 |
|---|---|---|

Schriftart: (Standard) Times New Roman, Schriftfarbe: Automatisch

| Seite 15: [188] Formatiert | Nele | 16.08.21 18:50:00 |
|---|---|---|

Schriftart: (Standard) Times New Roman, Schriftfarbe: Automatisch

| Seite 15: [189] Formatiert | Nele | 16.08.21 18:50:00 |
|---|---|---|

Schriftart: (Standard) Times New Roman

| Seite 15: [190] Formatiert | Nele | 16.08.21 18:50:00 |
|---|---|---|

Schriftart: (Standard) Times New Roman

| Seite 15: [191] Formatiert | Nele | 16.08.21 18:50:00 |
|---|---|---|

Schriftart: (Standard) Times New Roman

| Seite 15: [192] Formatiert | Nele | 16.08.21 18:50:00 |
|---|---|---|

Schriftart: (Standard) Times New Roman

| Seite 15: [193] Formatiert | Nele | 16.08.21 18:50:00 |
|---|---|---|

Schriftart: (Standard) Times New Roman

| Seite 15: [194] Formatiert | Nele | 16.08.21 18:50:00 |
|---|---|---|

Schriftart: (Standard) Times New Roman

| Seite 15: [195] Formatiert | Nele | 16.08.21 18:50:00 |
|---|---|---|

Schriftart: (Standard) Times New Roman

| Seite 15: [195] Formatiert | Nele | 16.08.21 18:50:00 |
|---|---|---|

Schriftart: (Standard) Times New Roman

| Seite 15: [196] Formatiert | Nele | 16.08.21 18:50:00 |
|---|---|---|

Schriftart: (Standard) Times New Roman

| Seite 15: [197] Formatiert | Nele | 16.08.21 18:50:00 |
|---|---|---|

Schriftart: (Standard) Times New Roman

| Seite 15: [198] Formatiert | Nele | 16.08.21 18:50:00 |
|---|---|---|

Standard, Block, Einzug: Vor:  1 cm, Hängend:  1 cm, Zeilenabstand:  Doppelt,  Keine Aufzählungen oder Nummerierungen

| Seite 15: [199] Formatiert | Nele | 16.08.21 18:50:00 |
|---|---|---|

Schriftart: (Standard) Times New Roman

| Seite 15: [200] Gelöscht | Nele | 16.08.21 18:50:00 |
|---|---|---|

| Seite 15: [201] Formatiert | Nele | 16.08.21 18:50:00 |
|---|---|---|

Schriftart: (Standard) Times New Roman

| Seite 15: [202] Formatiert | Nele | 16.08.21 18:50:00 |
|---|---|---|

Schriftart: (Standard) Times New Roman

| Seite 15: [203] Formatiert | Nele | 16.08.21 18:50:00 |
|---|---|---|

Schriftart: (Standard) Times New Roman

| Seite 15: [204] Formatiert | Nele | 16.08.21 18:50:00 |
|---|---|---|

Schriftart: (Standard) Times New Roman

| Seite 15: [204] Formatiert | Nele | 16.08.21 18:50:00 |
|---|---|---|

Schriftart: (Standard) Times New Roman

| Seite 15: [204] Formatiert | Nele | 16.08.21 18:50:00 |
|---|---|---|

Schriftart: (Standard) Times New Roman

| Seite 15: [205] Formatiert | Nele | 16.08.21 18:50:00 |
|---|---|---|

Schriftart: (Standard) Times New Roman

| Seite 15: [206] Formatiert | Nele | 16.08.21 18:50:00 |
|---|---|---|

Schriftart: (Standard) Times New Roman

| Seite 15: [207] Formatiert | Nele | 16.08.21 18:50:00 |
|---|---|---|

Schriftart: (Standard) Times New Roman

| Seite 15: [208] Formatiert | Nele | 16.08.21 18:50:00 |
|---|---|---|

Schriftart: (Standard) Times New Roman

| Seite 15: [209] Formatiert | Nele | 16.08.21 18:50:00 |
|---|---|---|

Schriftart: (Standard) Times New Roman

| Seite 17: [210] Formatiert | Nele | 16.08.21 18:50:00 |
|---|---|---|

Schriftart: (Standard) Times New Roman, Schriftfarbe: Automatisch

| Seite 17: [211] Formatiert | Nele | 16.08.21 18:50:00 |
|---|---|---|

Schriftart: (Standard) Times New Roman, Schriftfarbe: Automatisch

| Seite 17: [212] Formatiert | Nele | 16.08.21 18:50:00 |
|---|---|---|

Schriftart: (Standard) Times New Roman

| Seite 17: [212] Formatiert | Nele | 16.08.21 18:50:00 |
|---|---|---|

Schriftart: (Standard) Times New Roman

| Seite 17: [213] Formatiert | Nele | 16.08.21 18:50:00 |
|---|---|---|

Schriftart: (Standard) Times New Roman, Schriftfarbe: Automatisch, Englisch (Vereinigtes Königreich)

| Seite 17: [214] Formatiert | Nele | 16.08.21 18:50:00 |
|---|---|---|

Schriftart: (Standard) Times New Roman

| Seite 17: [214] Formatiert | Nele | 16.08.21 18:50:00 |
|---|---|---|

Schriftart: (Standard) Times New Roman

| Seite 17: [215] Formatiert | Nele | 16.08.21 18:50:00 |
|---|---|---|

Schriftart: (Standard) Times New Roman

| Seite 17: [215] Formatiert | Nele | 16.08.21 18:50:00 |
|---|---|---|

Schriftart: (Standard) Times New Roman

| Seite 17: [216] Gelöscht | Nele | 16.08.21 18:50:00 |
|---|---|---|

| Seite 17: [217] Formatiert | Nele | 16.08.21 18:50:00 |
|---|---|---|

Schriftart: (Standard) Times New Roman, Schriftfarbe: Automatisch

| Seite 17: [218] Formatiert | Nele | 16.08.21 18:50:00 |
|---|---|---|

Schriftart: (Standard) Times New Roman, Fett, Schriftfarbe: Rot

| Seite 32: [219] Formatiert | Nele | 16.08.21 18:50:00 |
|---|---|---|

Nach: 0.63 cm

| Seite 20: [220] Formatiert | Nele | 16.08.21 18:50:00 |
|---|---|---|

Schriftart: (Standard) Times New Roman

| Seite 20: [220] Formatiert | Nele | 16.08.21 18:50:00 |
|---|---|---|

Schriftart: (Standard) Times New Roman

| Seite 20: [221] Formatiert | Nele | 16.08.21 18:50:00 |
|---|---|---|

Block, Einzug: Vor: 1 cm, Hängend: 1 cm, Zeilenabstand: Doppelt

| Seite 20: [222] Formatiert | Nele | 16.08.21 18:50:00 |
|---|---|---|

Schriftart: (Standard) Times New Roman

| Seite 20: [223] Formatiert | Nele | 16.08.21 18:50:00 |
|---|---|---|

Schriftart: (Standard) Times New Roman, Tiefgestellt

| Seite 20: [223] Formatiert | Nele | 16.08.21 18:50:00 |
|---|---|---|

Schriftart: (Standard) Times New Roman, Tiefgestellt

| Seite 20: [224] Formatiert | Nele | 16.08.21 18:50:00 |
|---|---|---|

Schriftart: (Standard) Times New Roman

| Seite 20: [225] Gelöscht | Nele | 16.08.21 18:50:00 |
|---|---|---|

| Seite 20: [226] Formatiert | Nele | 16.08.21 18:50:00 |
|---|---|---|

Schriftart: (Standard) Times New Roman

| Seite 20: [227] Gelöscht | Nele | 16.08.21 18:50:00 |
|---|---|---|

| Seite 20: [228] Formatiert | Nele | 16.08.21 18:50:00 |
|---|---|---|

Schriftart: (Standard) Times New Roman

| Seite 20: [229] Formatiert | Nele | 16.08.21 18:50:00 |
|---|---|---|

Block, Zeilenabstand:  Doppelt, Nicht vom nächsten Absatz trennen

| Seite 20: [230] Formatiert | Nele | 16.08.21 18:50:00 |
|---|---|---|

Schriftart: (Standard) Times New Roman

| Seite 20: [230] Formatiert | Nele | 16.08.21 18:50:00 |
|---|---|---|

Schriftart: (Standard) Times New Roman

| Seite 20: [230] Formatiert | Nele | 16.08.21 18:50:00 |
|---|---|---|

Schriftart: (Standard) Times New Roman

| Seite 20: [231] Formatiert | Nele | 16.08.21 18:50:00 |
|---|---|---|

Schriftart: (Standard) Times New Roman, Schriftfarbe: Automatisch, Englisch (Vereinigtes Königreich)

| Seite 20: [231] Formatiert | Nele | 16.08.21 18:50:00 |
|---|---|---|

Schriftart: (Standard) Times New Roman, Schriftfarbe: Automatisch, Englisch (Vereinigtes Königreich)

| Seite 20: [231] Formatiert | Nele | 16.08.21 18:50:00 |
|---|---|---|

Schriftart: (Standard) Times New Roman, Schriftfarbe: Automatisch, Englisch (Vereinigtes Königreich)

| Seite 20: [232] Gelöscht | Nele | 16.08.21 18:50:00 |
|---|---|---|

| Seite 20: [233] Formatiert | Nele | 16.08.21 18:50:00 |
|---|---|---|

Schriftart: (Standard) Times New Roman

| Seite 20: [234] Formatiert | Nele | 16.08.21 18:50:00 |
|---|---|---|

Schriftart: (Standard) Times New Roman

| Seite 20: [235] Formatiert | Nele | 16.08.21 18:50:00 |
|---|---|---|

Schriftart: (Standard) Times New Roman

| Seite 20: [235] Formatiert | Nele | 16.08.21 18:50:00 |
|---|---|---|

Schriftart: (Standard) Times New Roman

| Seite 20: [236] Gelöscht | Nele | 16.08.21 18:50:00 |
|---|---|---|

| Seite 20: [237] Formatiert | Nele | 16.08.21 18:50:00 |
|---|---|---|

Schriftart: (Standard) Times New Roman

| Seite 20: [237] Formatiert | Nele | 16.08.21 18:50:00 |
|---|---|---|

Schriftart: (Standard) Times New Roman

| Seite 20: [238] Formatiert | Nele | 16.08.21 18:50:00 |
|---|---|---|

Schriftart: (Standard) Times New Roman

| Seite 20: [239] Formatiert | Nele | 16.08.21 18:50:00 |
|---|---|---|

Schriftart: (Standard) Times New Roman

| Seite 20: [239] Formatiert | Nele | 16.08.21 18:50:00 |
|---|---|---|

Schriftart: (Standard) Times New Roman

| Seite 20: [240] Formatiert | Nele | 16.08.21 18:50:00 |
|---|---|---|

Schriftart: (Standard) Times New Roman

| Seite 20: [241] Formatiert | Nele | 16.08.21 18:50:00 |
|---|---|---|

Schriftart: (Standard) Times New Roman

| Seite 20: [242] Formatiert | Nele | 16.08.21 18:50:00 |
|---|---|---|

Schriftart: (Standard) Times New Roman

| Seite 20: [242] Formatiert | Nele | 16.08.21 18:50:00 |
|---|---|---|

Schriftart: (Standard) Times New Roman

| Seite 20: [243] Formatiert | Nele | 16.08.21 18:50:00 |
|---|---|---|

Schriftart: (Standard) Times New Roman

| Seite 20: [244] Formatiert | Nele | 16.08.21 18:50:00 |
|---|---|---|

Schriftart: (Standard) Times New Roman

| Seite 20: [245] Formatiert | Nele | 16.08.21 18:50:00 |
|---|---|---|

Schriftart: (Standard) Times New Roman

| Seite 20: [245] Formatiert | Nele | 16.08.21 18:50:00 |
|---|---|---|

Schriftart: (Standard) Times New Roman

| Seite 20: [245] Formatiert | Nele | 16.08.21 18:50:00 |
|---|---|---|

Schriftart: (Standard) Times New Roman

| Seite 20: [246] Formatiert | Nele | 16.08.21 18:50:00 |
|---|---|---|

Schriftart: (Standard) Times New Roman

| Seite 20: [247] Formatiert | Nele | 16.08.21 18:50:00 |
|---|---|---|

Schriftart: (Standard) Times New Roman

| Seite 20: [248] Formatiert | Nele | 16.08.21 18:50:00 |
|---|---|---|

Schriftart: (Standard) Times New Roman

| Seite 20: [249] Formatiert | Nele | 16.08.21 18:50:00 |
|---|---|---|

Schriftart: (Standard) Times New Roman

| Seite 20: [250] Formatiert | Nele | 16.08.21 18:50:00 |
|---|---|---|

Schriftart: (Standard) Times New Roman

| Seite 24: [251] Gelöscht | Nele | 16.08.21 18:50:00 |
|---|---|---|

| Seite 24: [252] Formatiert | Nele | 16.08.21 18:50:00 |
|---|---|---|

Schriftart: (Standard) Times New Roman, Schriftfarbe: Automatisch

| Seite 24: [253] Gelöscht | Nele | 16.08.21 18:50:00 |
|---|---|---|

| Seite 24: [254] Formatiert | Nele | 16.08.21 18:50:00 |
|---|---|---|

Schriftart: (Standard) Times New Roman, Schriftfarbe: Automatisch

| Seite 24: [255] Formatiert | Nele | 16.08.21 18:50:00 |
|---|---|---|

Schriftart: (Standard) Times New Roman, Schriftfarbe: Automatisch

| Seite 24: [256] Formatiert | Nele | 16.08.21 18:50:00 |
|---|---|---|

Schriftart: (Standard) Times New Roman, Schriftfarbe: Automatisch

| Seite 24: [257] Formatiert | Nele | 16.08.21 18:50:00 |
|---|---|---|

Schriftart: (Standard) Times New Roman, Schriftfarbe: Automatisch

| Seite 24: [258] Formatiert | Nele | 16.08.21 18:50:00 |
|---|---|---|

Schriftart: (Standard) Times New Roman, Schriftfarbe: Automatisch

| Seite 24: [259] Formatiert | Nele | 16.08.21 18:50:00 |
|---|---|---|

Schriftart: (Standard) Times New Roman, Schriftfarbe: Automatisch

| Seite 24: [260] Formatiert | Nele | 16.08.21 18:50:00 |
|---|---|---|

Schriftart: (Standard) Times New Roman, Schriftfarbe: Automatisch

| Seite 26: [261] Gelöscht | Nele | 16.08.21 18:50:00 |
|---|---|---|

| Seite 26: [262] Gelöscht | Nele | 16.08.21 18:50:00 |
|---|---|---|

| Seite 28: [263] Gelöscht | Nele | 16.08.21 18:50:00 |
|---|---|---|

| Seite 28: [264] Gelöscht | Nele | 16.08.21 18:50:00 |
|---|---|---|

| Seite 28: [265] Formatiert | Nele | 16.08.21 18:50:00 |
|---|---|---|

Schriftart: (Standard) Times New Roman, Schriftfarbe: Automatisch

| Seite 28: [266] Gelöscht | Nele | 16.08.21 18:50:00 |
|---|---|---|

| Seite 28: [267] Formatiert | Nele | 16.08.21 18:50:00 |
| --- | --- | --- |

Schriftart: (Standard) Times New Roman, Nicht Fett

| Seite 28: [268] Gelöscht | Nele | 16.08.21 18:50:00 |
| --- | --- | --- |

| Seite 28: [269] Formatiert | Nele | 16.08.21 18:50:00 |
| --- | --- | --- |

Schriftart: (Standard) Times New Roman, Schriftfarbe: Automatisch

| Seite 29: [270] Gelöscht | Nele | 16.08.21 18:50:00 |
| --- | --- | --- |

| Seite 29: [271] Gelöscht | Nele | 16.08.21 18:50:00 |
| --- | --- | --- |

| Seite 29: [272] Formatiert | Nele | 16.08.21 18:50:00 |
| --- | --- | --- |

Block, Einzug: Erste Zeile:  0 cm, Zeilenabstand:  Doppelt

| Seite 29: [273] Formatiert | Nele | 16.08.21 18:50:00 |
| --- | --- | --- |

Schriftart: (Standard) Times New Roman, Schriftfarbe: Automatisch

| Seite 29: [274] Formatiert | Nele | 16.08.21 18:50:00 |
| --- | --- | --- |

Schriftart: (Standard) Times New Roman, Schriftfarbe: Automatisch

| Seite 29: [275] Formatiert | Nele | 16.08.21 18:50:00 |
| --- | --- | --- |

Schriftart: (Standard) Times New Roman, Schriftfarbe: Automatisch

| Seite 30: [276] Formatiert | Nele | 16.08.21 18:50:00 |
| --- | --- | --- |

Schriftart: (Standard) Times New Roman, Schriftfarbe: Automatisch

| Seite 30: [276] Formatiert | Nele | 16.08.21 18:50:00 |
| --- | --- | --- |

Schriftart: (Standard) Times New Roman, Schriftfarbe: Automatisch

| Seite 30: [276] Formatiert | Nele | 16.08.21 18:50:00 |
| --- | --- | --- |

Schriftart: (Standard) Times New Roman, Schriftfarbe: Automatisch

| Seite 30: [277] Formatiert | Nele | 16.08.21 18:50:00 |
| --- | --- | --- |

Schriftart: (Standard) Times New Roman, Schriftfarbe: Automatisch

| Seite 30: [277] Formatiert | Nele | 16.08.21 18:50:00 |
| --- | --- | --- |

Schriftart: (Standard) Times New Roman, Schriftfarbe: Automatisch

| Seite 30: [278] Formatiert | Nele | 16.08.21 18:50:00 |
| --- | --- | --- |

Schriftart: (Standard) Times New Roman

| Seite 30: [278] Formatiert | Nele | 16.08.21 18:50:00 |
| --- | --- | --- |

Schriftart: (Standard) Times New Roman

| Seite 30: [279] Formatiert | Nele | 16.08.21 18:50:00 |
| --- | --- | --- |

Schriftart: (Standard) Times New Roman

| Seite 30: [279] Formatiert | Nele | 16.08.21 18:50:00 |
| --- | --- | --- |

Schriftart: (Standard) Times New Roman

| Seite 30: [279] Formatiert | Nele | 16.08.21 18:50:00 |
| --- | --- | --- |

Schriftart: (Standard) Times New Roman

| Seite 30: [279] Formatiert | Nele | 16.08.21 18:50:00 |
|---|---|---|

Schriftart: (Standard) Times New Roman

| Seite 30: [279] Formatiert | Nele | 16.08.21 18:50:00 |
|---|---|---|

Schriftart: (Standard) Times New Roman

| Seite 30: [280] Formatiert | Nele | 16.08.21 18:50:00 |
|---|---|---|

Schriftart: (Standard) Times New Roman, Englisch (Vereinigtes Königreich)

| Seite 30: [280] Formatiert | Nele | 16.08.21 18:50:00 |
|---|---|---|

Schriftart: (Standard) Times New Roman, Englisch (Vereinigtes Königreich)

| Seite 30: [281] Gelöscht | Nele | 16.08.21 18:50:00 |
|---|---|---|

| Seite 30: [281] Gelöscht | Nele | 16.08.21 18:50:00 |
|---|---|---|

| Seite 30: [281] Gelöscht | Nele | 16.08.21 18:50:00 |
|---|---|---|

| Seite 31: [282] Gelöscht | Nele | 16.08.21 18:50:00 |
|---|---|---|